# Preparation of oxygen-sensitive proteins for high-resolution cryoEM structure determination using blot-free vitrification

Brian D. Cook [1,2], Sarah M. Narehood [1,2], Kelly L. McGuire [1,2], Yizhou Li [1], F. Akif Tezcan [1] & Mark A. Herzik Jr [1] ✉

High-quality grid preparation for single-particle cryogenic electron microscopy (cryoEM) remains a bottleneck for routinely obtaining high-resolution structures. The issues that arise from traditional grid preparation workflows are particularly exacerbated for oxygen-sensitive proteins, including metalloproteins, whereby oxygen-induced damage and alteration of oxidation states can result in protein inactivation, denaturation, and/or aggregation. Indeed, 99% of the current structures in the EMBD were prepared aerobically and limited successes for anaerobic cryoEM grid preparation exist. Current practices for anaerobic grid preparation involve a vitrification device located in an anoxic chamber, which presents significant challenges including temperature and humidity control, optimization of freezing conditions, costs for purchase and operation, as well as accessibility. Here, we present a streamlined approach that allows for the vitrification of oxygen-sensitive proteins in reduced states using an automated blot-free grid vitrification device – the SPT Labtech chameleon. This robust workflow allows for high-resolution structure determination of dynamic, oxygen-sensitive proteins, of varying complexity and molecular weight.

Metalloproteins are critically important for numerous biological processes, including electron transport in mitochondria, photosynthesis, and nitrogen fixation, and understanding their structures and functions remains critically important[1–3]. A large subset of these vital protein complexes harbor metal clusters that are susceptible to damage or altered function upon exposure to molecular oxygen ($O_2$)[4,5]. Structural characterization of these $O_2$-sensitive metalloproteins has been hampered because they require specialized anaerobic setups and methodologies to minimize the risk of potentially obtaining results that inaccurately represent their functional states[6,7]. Indeed, the determination of the structures of such metalloproteins using X-ray crystallography requires crystallogenesis to occur under anaerobic conditions, typically necessitating crystal trays to be in an anaerobic chamber for growth and harvesting[8]. Although there have been many successes using such approaches[9–14], the conditions needed for

crystallization often preclude visualizing these complexes under catalytic turnover conditions due to the inherent dynamics involved in such processes and the timescales for crystal growth. As has been recently shown for several proteins, including the metalloenzyme nitrogenase, cryogenic electron microscopy (cryoEM) is well-suited to capturing $O_2$-sensitive metalloproteins under catalytic turnover conditions[15–18], provided that the significant challenges for the preparation of cryoEM samples in anaerobic conditions can be overcome.

In the past decade, significant technological advances in microscope hardware and data processing algorithms have established cryoEM as a leading technique for macromolecular structure determination[19]. While these technological advances continue to pay dividends with near-exponential growth in the number of structures determined by cryoEM each year, sample preparation remains a key bottleneck[20,21]. The unpredictable nature of cryoEM sample

[1]Department of Chemistry and Biochemistry, University of California San Diego, La Jolla, USA. [2]These authors contributed equally: Brian D. Cook, Sarah M. Narehood, Kelly L. McGuire. ✉e-mail: mherzik@ucsd.edu

vitrification frequently results in partial or complete destabilization of samples, preferential orientation, or inconsistent freezing behavior[20,22–24]. Notably, these issues are exacerbated in the case of $O_2$-sensitive samples that require anaerobic handling conditions, as $O_2$ exposure can result in oxidation and/or loss of metal(s)/cofactor(s), loss of activity, and even protein denaturation[4,25–27]. To date, cryoEM structure determination of anaerobic proteins has relied on custom sample preparation setups and/or specialized equipment[15,16]. Indeed, to address the multitude of challenges associated with preparing cryoEM grids anaerobically, several recent studies have approached these challenges by conducting this process in an anoxic environment, such as an anaerobic chamber, or by protecting the sample from $O_2$ once relocated to an aerobic environment[15,16]. While dedicating a cryoEM grid preparation device within an anaerobic chamber is an effective strategy to limit or eliminate $O_2$ during cryoEM grid preparation, this approach is costly, and lack of accessibility prevents widespread adoption across the community. Establishing a standardized workflow for freezing $O_2$-sensitive proteins using grid preparation devices readily available at cryoEM facilities would facilitate broadscale adoption in a critical area of structural biology.

Here, we present a workflow that utilizes the SPT Labtech chameleon – a next-generation nanospray cryoEM grid preparation device – in an aerobic environment to freeze $O_2$-sensitive proteins under functionally anaerobic conditions, which we term (an)aerobic. Our workflow maintains (an)aerobic samples by protecting against or eliminating $O_2$ contamination prior to loading into the chameleon, during sample aspiration, and throughout sample deposition on self-wicking EM grids, respectively. Demonstrating the utility of this workflow, we determined high-resolution cryoEM structures of deoxygenated human hemoglobin (Hb) in various liganded states and the molybdenum-iron protein (MoFeP) – the $O_2$-sensitive catalytic component of *Azotobacter vinelandii* nitrogenase – in a reduced state. Finally, we detail the effects of $O_2$ contamination at different stages of the workflow and show how to prevent their effects. Together, we demonstrate reproducible high-resolution structure determination of $O_2$-sensitive proteins of varying complexity and size – as small as

64 kDa – with a workflow requiring only minimal modifications to a robust, widely available EM grid freezing method.

## Results
### Development of a specialized workflow for freezing (an)aerobic samples using the SPT Labtech chameleon

While standard cryoEM grid preparation by blot-and-plunge freezing has remained essentially unchanged for decades, a new generation of sample preparation devices uses nanoliter sample dispensing heads and self-wicking cryoEM grids to automate sample preparation. These devices, including Spotiton and the SPT Labtech chameleon, enable precise control over ice thickness and also potentially address air-water interface issues and preferred orientation problems[28–33]. This vitrification method has the potential to enhance and modernize current freezing practices for studying challenging proteins and has been demonstrated to enable freezing of unstable or sensitive proteins[33–35]. We sought to adapt the aerobic chameleon grid vitrification protocol for anaerobic samples by systematically addressing key stages of the workflow where $O_2$ could contaminate the sample. These stages include: (1) pre-loading, during which the sample is exposed to $O_2$ before loading into the chameleon's piezo-electric dispenser, (2) pre-deposition, during which $O_2$ may enters the piezo-dispenser before sample deposition onto the EM grid, and (3) post-deposition, where $O_2$ exposure may occur during the auto-wicking process on the EM grid prior to vitrification.

We chose an $O_2$-binding protein, human hemoglobin (Hb), as a model system for development of an (an)aerobic workflow for sample preparation in the chameleon (Fig. 1A). Briefly, Hb is a heterotetramer ($\alpha_2\beta_2$) with each subunit harboring a heme cofactor that is capable of binding $O_2$ in solution when the heme iron is in a ferrous ($Fe^{2+}$) state[36,37]. Importantly, the oxidation and ligand-bound states of Hb can be readily monitored using UV-Vis spectroscopy (Fig. 1B) and further verified using cryoEM[38–40]. We first elected to limit the amount of $O_2$ that enters the sample prior to loading in the chameleon dispenser. Briefly, to prepare EM grids using the chameleon, the isolated protein sample must first be transferred into a dedicated single-use chameleon

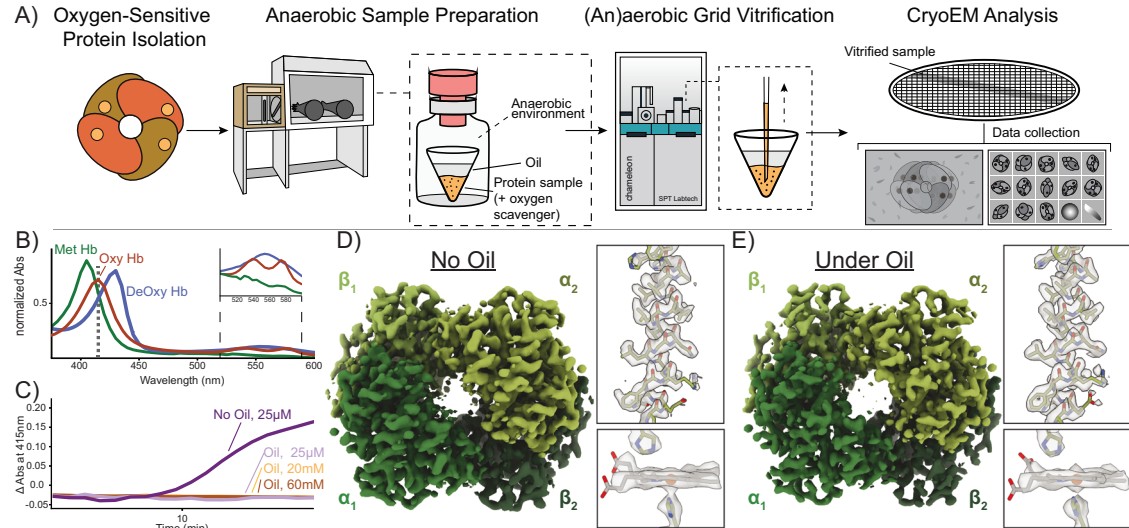

**Fig. 1 | Development of the (an)aerobic SPT Labtech chameleon grid vitrification protocol. A** Schematic of the (an)aerobic cryoEM grid vitrification protocol using the SPT Labtech chameleon. In an anaerobic environment, the $O_2$-sensitive sample is placed into the chameleon sample cup and then covered with a layer of anaerobic Al's oil to protect the sample from $O_2$. The sample cup is then transferred into a glass vial and sealed for transport to the chameleon for preparation of high-quality grids. **B** UV-Vis analysis of the different human Hb oxidation and liganded states discussed in this study. Methemoglobin (metHb, green line), deoxyHb (blue

line), and oxyHb (red line) have distinct Soret peak maximums at 405 nm, 430 nm, and 415 nm, respectively. The dashed line represents abs (415 nm) monitored in (**C**). **C** $O_2$ titration curves of deoxyHb with varying NaDT concentrations and with or without the presence of Al's oil. **D, E** Comparison of metHb densities determined with or without the protective Al's oil layer detailing no obvious differences in the metHb densities between the sample with oil present or not. For each structure, zoomed-in views of $\alpha$ helix 100–118 and the heme cofactors of subunit $\alpha_1$ detail the high quality of both reconstructions.

sample cup and then placed into the instrument for aspiration into the dispenser. For aerobic samples, this process can readily be performed at the instrument immediately prior to vitrification[34]. To achieve this same process for anaerobic samples, there needs to be a mechanism to prevent or limit $O_2$ perfusion into the sample before aspiration. To achieve this, we elected to add a protective $O_2$ barrier containing a 50:50 mixture of paraffin oil and silicon oil (a.k.a. Al's oil) directly on top of the sample (Fig. 1A, C, and Supplementary Fig. S1)[41–43]. Due to the physical properties of the oil layer, $O_2$ perfusion into the anaerobic sample should be slowed and, therefore, help to maintain anaerobic conditions until sample aspiration[44]. To verify this, we prepared deoxygenated Hb by resuspending lyophilized human methemoglobin (metHb) in 1x phosphate-buffered saline (PBS) and then reducing the heme irons in metHb from the ferric ($Fe^{3+}$) state to the ferrous ($Fe^{2+}$) state (deoxygenated Hb; herein referenced as deoxyHb) using the chemical reductant, sodium dithionite (NaDT), in an anaerobic chamber. As monitored by UV-Vis spectroscopy, metHb has a Soret maximum at 405 nm, indicating that the heme iron is in an oxidized state ($Fe^{3+}$) and incapable of binding $O_2$. When the iron centers are reduced ($Fe^{2+}$) using NaDT, a noticeable shift in the Soret maximum to ~ 430 nm occurs, while a broad peak is detected at 550 nm (Fig. 1B)[45]. We then placed 1 ml of deoxyHb into a cuvette and added anaerobic Al's oil on top to the same approximate height of the chameleon sample cup (~ 5 mm). We then monitored $O_2$ binding to deoxyHb by measuring the change in absorbance at 415 nm – the Soret maximum of $O_2$-bound Hb (oxyHb) – following exposure to $O_2$ (Fig. 1B, C, and Supplementary Fig. S2)[45]. Notably, this sample was able to remain deoxygenated under air for up to 20 min. However, when compared to the same deoxyHb sample without the protective oil layer, $O_2$ binding occurred much more rapidly, indicating that the oil layer is a sufficient first line of protection (Fig. 1C and Supplementary Fig. S2). Although these conditions do not perfectly mimic those the sample will experience in the chameleon sample cup, the extent of protection that the anaerobic oil layer provides should be similar.

During sample aspiration, the chameleon dispenser needs to penetrate through the protective oil layer to access the sample at the bottom of the chameleon sample cup (Fig. 1A). Immediate inspection of the dispenser tip in the chameleon software after this step indicates that residual oil remains on the outside of the tip, likely affecting the dispensing efficiency (Supplementary Fig. S1). To remedy this problem, we incorporated additional dispenser wash steps to remove any residual oil and restore normal dispenser behavior (Supplementary Fig. S1 and **Methods**). Although these steps can be performed quickly, we wanted to ensure that this would not lead to potential issues in dispenser operation, sample deposition on the EM grid, and/or a decrease in sample integrity. To eliminate these possibilities, we next assessed the effect of the protective oil layer on grid preparation and downstream cryoEM imaging and data processing using our metHb sample. Briefly, we created two metHb samples – one without and one with the protective oil layer – by placing 15 μL of metHb (~ 125 μM) each into two sample cups. One sample was used without modification (non-oiled), and ~ 70 μL of aerobic Al's oil was carefully layered over the second sample up to the brim of the sample cup (oiled). Two cryoEM grids of each sample were prepared using the chameleon following recommended guidelines[34]. Both metHb samples were imaged using the same microscope parameters with approximately the same number of micrographs for each dataset collected (see **Methods**). We did not observe any noticeable differences neither in the quality of the sample stripes on the grids nor in their behavior during data collection. Both datasets were processed independently but with similar steps and parameters to allow for a direct comparison of the resulting densities (Supplementary Fig. S3). A comparison of our 2.78-Å resolution non-oiled and 2.71-Å resolution oiled metHb EM densities did not indicate any notable differences in their overall quality (Fig. 1D, E). Indeed, the view distribution and the total number of particles were

similar between the samples, each resolved to a similar resolution and demonstrated similar map-to-model validation metrics (Fig. 1D, E, Supplementary Fig. S3, and Supplementary Table S1). Thus, the addition of Al's oil to protect the sample during chameleon grid preparation does not significantly hinder the operation of the chameleon nor our ability to obtain high-resolution structures.

## CryoEM structures of oxy, mixed, and deoxy hemoglobin to assess the (an)aerobic chameleon workflow

A major source of potential $O_2$ contamination is the transportation from the anaerobic chamber to the chameleon. To address this issue, we placed the oil-layered sample inside a gas-tight glass vial (e.g., Reacti-vial) and sealed it with a rubber septum inside the anaerobic chamber to preserve the anoxic environment. Notably, the samples must remain stored in this condition until the chameleon is ready for sample aspiration. Once the chameleon is ready for sample loading, the septum is removed, and the chameleon sample cup containing the protective oil-layered sample is carefully transferred from the glass vial to the chameleon sample cup holder and immediately aspirated into the chameleon dispenser (Fig. 1A and Supplementary Fig. S1). EM grid preparation can then be performed normally (Fig. 1A)[34]. We surmised that the oil layer alone would not be sufficient to eliminate $O_2$ introduction into the sample, so we decided to incorporate additional measures to eliminate $O_2$ perfusion into the sample during grid freezing and determine the structure of deoxyHb as our $O_2$ sensor. Specifically, we sought to address potential $O_2$ contamination in our samples during the vitrification process by simultaneously optimizing the critical 'wicking period' following sample deposition onto the grid necessary for sufficiently thin ice in our cryoEM samples and the necessary concentration of NaDT – a versatile reducing agent that is also capable of depleting solutions of dissolved $O_2$ – in the sample buffer required to maintain anaerobic conditions[46]. Importantly, the wicking time is a key variable during grid preparation using the chameleon as this phase significantly influences the uniformity and thinness of the resultant ice layer and can be easily tuned by the user: too short and the ice will be too thick for ideal imaging, too long and this will lead to complete oxygenation of the sample and potentially yield too thin an ice layer. By balancing the NaDT concentrations and the wicking time, we sought to determine the cryoEM structure of deoxyHb.

To begin to explore the relationship between [NaDT] and wicking time, we first prepared deoxyHb in PBS buffer supplemented with 25 μM NaDT (here termed "very low NaDT"). Although this concentration of NaDT is an order of magnitude below the predicted concentration of $O_2$ (250 μM) in solutions prepared under an ambient atmosphere, we included it to ensure that all hemes remained in the ferrous state after initial preparation of deoxyHb and buffer exchange[47]. We then transferred the deoxyHb sample from the anaerobic environment to the chameleon using the protective oil layer conditions described above (Fig. 1A) and prepared EM grids using similar conditions and wicking times as our metHb samples. We then determined the structure of this "very low NaDT" Hb at a resolution of 2.55-Å (Fig. 2A, Supplementary Fig. S4, and Supplementary Table S1). This structure revealed that Hb was in the relaxed (R) conformation (as indicated by the rotation of $\alpha_1\beta_1$ in relation to the $\alpha_2\beta_2$), which is the form of Hb with a high affinity of $O_2$.[36] Evaluation of the heme pocket for each subunit indicated a density above each heme that was not present in our metHb structures, consistent with molecular $O_2$ binding to the Fe centers (Fig. 1D, E, 2A, and Supplementary Fig. S10). The $O_2$-heme coordination metrics and geometries closely correspond to those observed in the X-ray structures of oxyHb (Supplementary Fig. S5).[48] In addition, we used Phenix.resolve_cryo_em – a cryoEM density modification tool adapted from X-ray crystallography – to better visualize the $O_2$ density above the hemes (Fig. 2A)[49]. This dataset demonstrated that during sample deposition and self-wicking, $O_2$

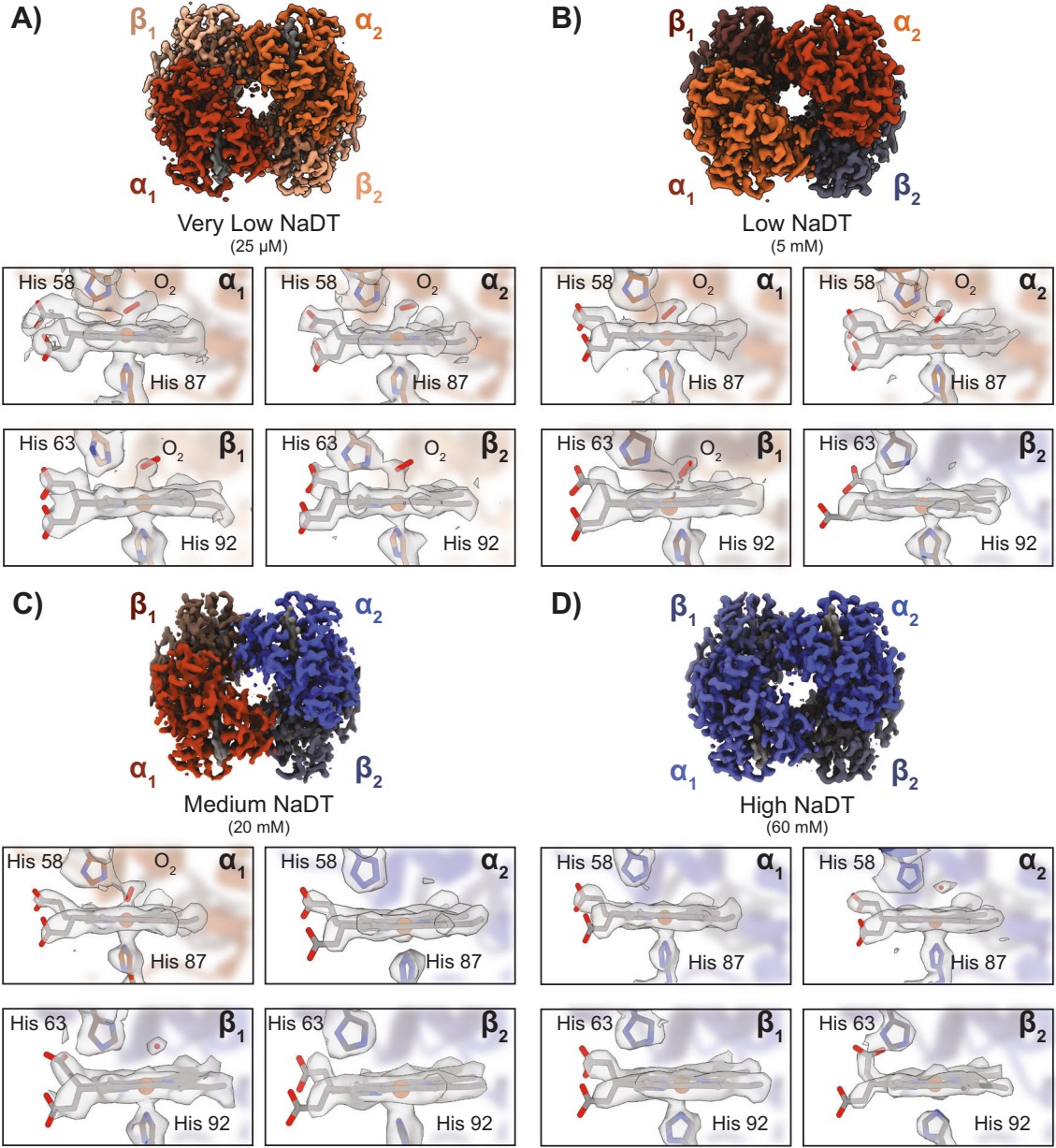

**Fig. 2 | CryoEM structures of various Hb species using the (an)aerobic chameleon protocol. A** 2.55-Å resolution cryoEM structure of oxyHb determined under 25 μM NaDT. Zoomed-in views (shown below) of the α1 and β1 heme cofactors indicate a clear density of bound O2 above the Fe of each heme. **B** 2.52-Å resolution cryoEM structure of oxyHb determined under 5 mM NaDT. Zoomed-in views (shown below) of the α1 and β1 heme cofactors indicate a clear density of bound O2 above the Fe of each heme. **C** 2.72-Å resolution cryoEM structure of a partially oxygenated Hb species obtained under 20 mM NaDT. Clear O2 density was found above the α1 heme, while the α2, β1, and β2 heme groups lacked any such density. **D** 2.75-Å resolution cryoEM structure of deoxyHb under 60 mM NaDT. Orange and red α and β subunits have oxygen present, while blue subunits are deoxygenated.

exchanges rapidly enough for Hb to bind it. Thus, determining a NaDT concentration sufficient to maintain anaerobic conditions is critical.

To find a concentration of NaDT that would protect our sample from O2 contamination while also allowing for proper biological function, we determined the structure of deoxy-Hb at different NaDT concentrations. Given that the concentration of dissolved O2 is approximately 250 μM, we used a NaDT concentration of 5 mM to potentially capture any O2 exchanging during sample deposition and wicking. Using similar conditions as the no-NaDT oxyHb, we generated a Hb sample under anerobic conditions supplemented with 5 mM

NaDT (termed "low NaDT") and used approximately the same wicking time as our no-NaDT oxyHb. We transported this sample under oil and in the gas-tight vial to the chameleon. Following grid preparation, we collected a dataset of this "low-NaDT" sample and determined a 2.52-Å resolution structure of Hb (Fig. 2B, Supplementary Fig. S6, and Supplementary Table S1). Interestingly, this Hb structure once again had similar densities of heme-bound O2's at three of the four hemes within the tetramer similar to the "very low NaDT" oxyHb sample. Surprisingly, there was no obvious O2 density above the β2 heme, implying that the NaDT concentration was sufficiently high to decrease the

dissolved $O_2$ in solution below the partial pressure needed to have occupancy at all four Hb hemes. In addition, the "low-NaDT" Hb tetramer was in the R state, and the refined models matched previous $O_2$-heme coordination geometries (Supplementary Fig. S5). Although the concentration of NaDT used in this sample was an order of magnitude higher than the theoretical dissolved $O_2$ concentration, we reasoned that as the liquid layer thins, there may be increased oxygen exchange. To prevent this, we decreased the sample wicking time while simultaneously increasing the NaDT concentration.

We next sought to decrease the wicking time used in our metHb/oxyHb structures (from ~ 900 ms to ~ 300 ms) to limit the total time the sample was exposed to air after deposition onto the grid, while also increasing the [NaDT] to 20 mM (termed "medium NaDT") to deplete any $O_2$ entering the sample. DeoxyHb was prepared similarly as described above and exchanged into PBS supplemented with 20 mM NaDT prior to transporting under oil and transfer to the chameleon. We prepared cryoEM grids using our modified (an)aerobic chameleon protocol (Fig. 1A and **Methods**) and collected a similarly-sized dataset as metHb/oxyHb to determine a 2.72-Å resolution structure of Hb (Fig. 2C, Supplementary Fig. S7, and Table S1). Remarkably, when compared to our oxyHb structure, the densities of the heme-bound $O_2$'s were weaker and displayed heterogeneity across the different hemes. Specifically, we observed that the $α_1$ heme showed bound $O_2$, while the $α_2$, $β_1$, and $β_2$ hemes lacked any such density (Fig. 2C and Supplementary Fig. S10). The densities in the $α_1$ subunit are similar in nature to those observed in our oxyHb structure and indicate that it is most likely molecular $O_2$ with the $α_2$, $β_1$, and $β_2$ hemes in the unliganded state (Supplementary Fig. S5 and S10).

The above data suggest that 20 mM NaDT provided some protective effect but did not completely eliminate $O_2$ perfusion into the sample during sample deposition and vitrification, despite the shortened wicking time used for this sample. As wicking time could not be further reduced without resulting in unacceptably thick ice for such a low molecular weight sample, we increased the final concentration of NaDT to 60 mM (termed "high NaDT") to ensure complete elimination of contaminating $O_2$. We prepared deoxyHb in the same manner described above in PBS supplemented with 60 mM NaDT in the final buffer and prepared cryoEM grids using identical conditions as the mixed-$O_2$ species (Supplementary Table S1). We collected a dataset of similar size and determined the structure Hb to a resolution of 2.75-Å (Fig. 2D, Supplementary Fig. S8, and Supplementary Table S1). Although the overall tetramer structure remained in the R state, our analysis of the heme groups in each subunit revealed an absence of any density extending above the iron atom. This lack of density indicates that no observable $O_2$ is bound to the hemes. In addition, when we fit a previously determined deoxyHb atomic model into the density, well-defined water found in the X-ray model fits into a density above the heme (Supplementary Fig. S9). We also compared the densities above each heme in our Hb structures to recently published cryoEM maps of both deoxy and oxy human Hb, as well as to two structures of Hb determined using X-ray crystallography, further confirming the ligand states of each of our structures (Supplementary Fig. S10)[48,50]. Together, these structures indicate that, in combination, the protective oil layer, minimized sample wicking times, and appropriate [NaDT] are sufficient to maintain deoxyHb until structure determination and, therefore, create sufficiently reducing conditions for $O_2$-sensitive samples.

### CryoEM structures of *A. vinelandii* MoFeP under (an)aerobic conditions

Having demonstrated the effectiveness of our protocol in (an)aerobically preparing cryoEM grids for the determination of deoxyHb structure using the chameleon, we next targeted the nitrogenase MoFeP – an $O_2$-sensitive metalloenzyme that plays a pivotal role in converting atmospheric dinitrogen ($N_2$) to ammonia ($NH_3$)[1,51]. Despite decades of research, there continue to be numerous outstanding

questions pertaining to the structural intermediates of biological $N_2$ fixation that remain unanswered due to limitations in structural techniques for investigating $O_2$-sensitive enzymes.

Nitrogenase comprises two subunits: the reductase subunit, known as iron protein (FeP), and the catalytic subunit, known as MoFeP. MoFeP is a heterotetramer ($α_2β_2$) composed of two symmetry-related αβ dimers, each containing an [8Fe:7S] electron-relay cluster (P-cluster) and a [7Fe:1Mo:9S:1C] catalytic cofactor (FeMoco). Due to the sensitivity of these metal clusters, most structural studies of nitrogenase have been performed through anaerobic X-ray crystallography[52–55]. While informative, X-ray crystallography is not ideally suited to capture transient catalytic states associated with large conformational changes, which is the case in nitrogenase. Recent studies of nitrogenase have used cryoEM to uncover important dynamic steps within the nitrogenase cycle[15,16]. However, these studies relied on freezing techniques that are either difficult to replicate or require costly/laborious implementation of grid preparation devices in an anaerobic environment. Thus, we decided to use MoFeP to determine if our (an)aerobic freezing protocol can be used on a highly $O_2$-sensitive sample.

To demonstrate the $O_2$ sensitivity of MoFeP during traditional aerobic grid vitrification, we first froze cryoEM grids of NaDT-reduced MoFeP using similar conditions as deoxyHb but without the protective oil layer or NaDT in the cryoEM sample buffer. Using our chameleon protocol, we prepared grids of MoFeP and obtained a 2.39-Å resolution structure (Fig. 3A, Supplementary Fig. S11, and Supplementary Table S2). As expected from previous studies of nitrogenase under conditions of $O_2$ exposure, we did not see any obvious changes or damage to FeMoco (Fig. 3B), but an inspection of the EM density around the P-cluster suggested that it was in the 2-electron oxidized $P^{2+}$ state, indicating $O_2$ perfusion through the sample and oxidation of the P-cluster from the $P^N$ reduced state to $P^{2+}$ (Fig. 3C). The P-cluster resembles two fused 4Fe:4S clusters and is responsible for electron transfer from the FeP to FeMoco during the catalytic cycle. However, the P-cluster is also highly sensitive to $O_2$ exposure and is rapidly oxidized under aerobic conditions, that results in significant alterations to the metal cluster geometry and coordination. When the fully reduced $P^N$ resting state of the P-cluster undergoes one-electron oxidization to the $P^{1+}$ state, the Fe6 atom that is initially coordinated to S1 in $P^N$ becomes coordinated to a neighboring serine within the beta subunit ($β^IS188$) (Fig. 3C; red arrow). A further one-electron oxidation of the cluster to the $P^{2+}$ state leads to the additional coordination of the Fe5 atom to the amide backbone of one of the coordinating cysteines in the alpha subunit ($α^IC88$), which is accompanied by the dissociation of the bond between Fe5 and the bridging S1 sulfur atom (Fig. 3C; green arrow). Both of these linkages are present in our aerobically-prepared MoFeP structure, confirming oxidation of the P-clusters to the $P^{2+}$ state following $O_2$ exposure during grid preparation (Fig. 3A and C).

We then prepared cryoEM grids of NaDT-reduced MoFeP using chameleon freezing conditions similar to those used to obtain our deoxyHb structure, a final [NaDT] of 60 mM (Fig. 3D, Supplementary Fig. S12, and Supplementary Table S2). Analysis of our 2.08-Å resolution anaerobic MoFeP cryoEM structure indicates an intact FeMoco (Fig. 3E). In addition, we found a fully reduced P-cluster $P^N$ was preserved with no evidence of density that would indicate the dissociation of the bonds to the bridging S1 or coordination with neighboring residues as observed in the $P^{1+}$ and $P^{2+}$ states (Fig. 3F, red and green arrows). While this concentration of NaDT protected MoFeP and its metal clusters from oxygen damage, the high concentration of NaDT may preclude functional activity for future studies. To this end, we explored whether the same reduced MoFeP state could be achieved at a [NaDT] of 20 mM. We prepared cryoEM grids in the same manner as the 60 mM NaDT dataset, however, in the presence of 20 mM NaDT. We determined a second reduced MoFeP structure at 2.08 Å resolution

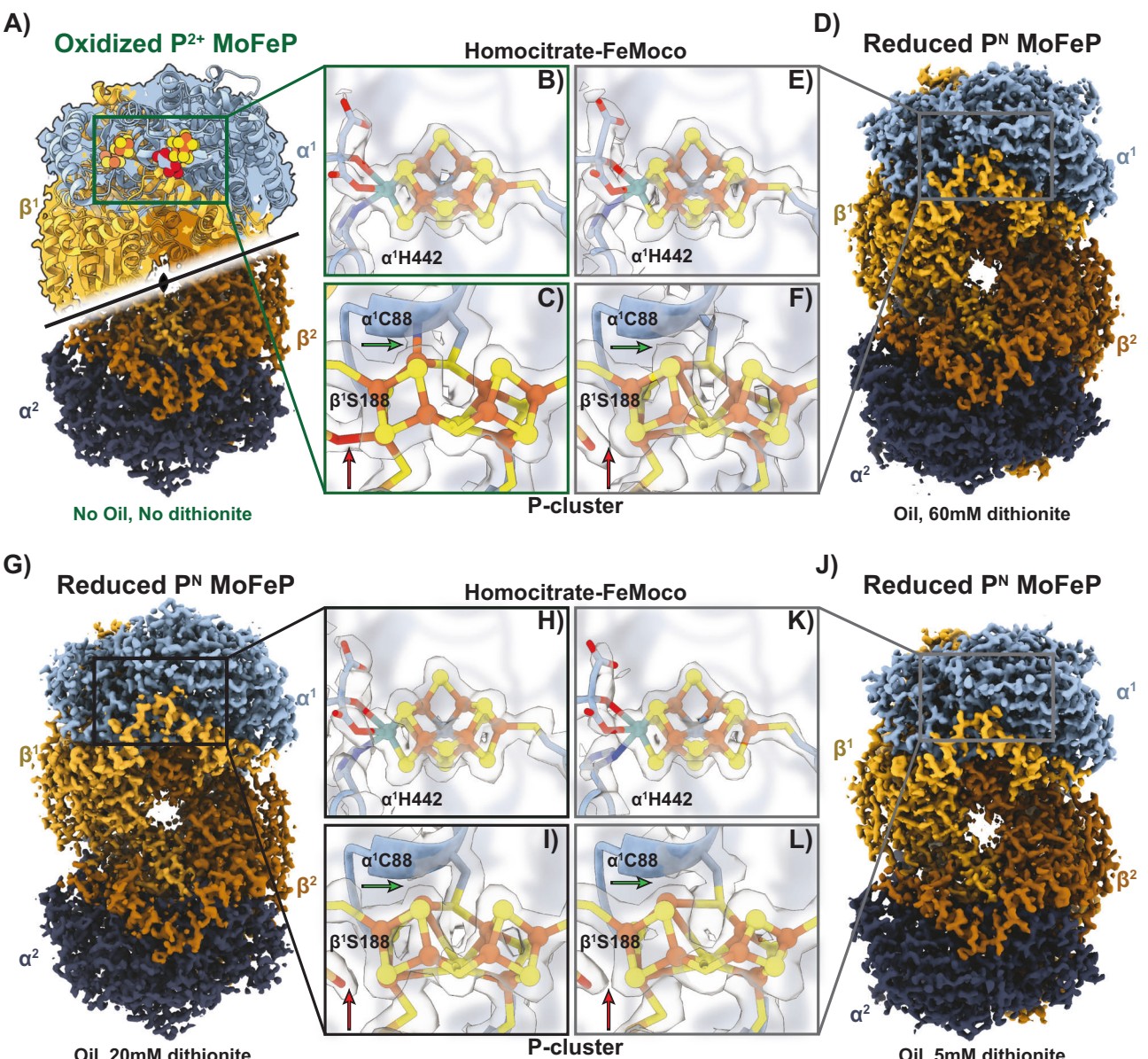

**Fig. 3 | CryoEM structures of oxidized and reduced MoFeP. A** 2.39-Å resolution cryoEM structure of oxidized $P^{2+}$ *Av*MoFeP obtained using the chameleon aerobically (without protective oil or NaDT). The $C_2$ symmetry axis is indicated. **B** The EM density for the homocitrate-FeMoco ligand of the oxidized MoFeP does not exhibit significant change or damage. **C** The oxidized P-cluster ($P^{2+}$ state) from the sample prepared aerobically (**A**) indicates clear density for linkages between the $\alpha^1$C88 amide backbone nitrogen with Fe5 (green arrow) and the $\beta^1$S88 side-chain with Fe6 (red arrow) of the P-cluster. **D** 2.08-Å resolution cryoEM structure of reduced $P^N$ AvMoFeP obtained using the (an)aerobic chameleon protocol with protective oil and 60 mM NaDT. **E** The homocitrate and FeMoco do not show significant changes in the reduced MoFeP structure. **F** The linkages indicative of $O_2$ damage are lacking in the (an)aerobically-prepared MoFeP, indicating a reduced $P^N$ P-cluster (red and green arrows). **G** 2.08-Å resolution cryoEM structure of reduced $P^N$ *Av*MoFeP obtained using the (an)aerobic chameleon protocol with protective oil and 20 mM NaDT. **H** The EM density for the homocitrate-FeMoco ligand of the reduced MoFeP does not exhibit significant change or damage. **I** The reduced $P^N$ P-cluster from the sample prepared anaerobically lacks clear density for linkages between the $\alpha^1$C88 amide backbone nitrogen with Fe5 (green arrow) and the $\beta^1$S88 side-chain with Fe6 (red arrow) of the P-cluster. **J** 2.19-Å resolution cryoEM structure of reduced ($P^N$) AvMoFeP obtained using the (an)aerobic chameleon protocol with protective oil and 5 mM NaDT. **K** Similar to (**B**), the homocitrate and FeMoco do not show significant changes in the reduced MoFeP structure. **L** The linkages indicative of $O_2$ damage are lacking in the (an)aerobically-prepared MoFeP, indicating a reduced $P^N$ P-cluster (red and green arrows). **B**, **C**, **E**, **F**, **H**, **I**, **K**, **L**) Surrounding residues were rendered transparent for clarity.

and analyzed the same metal clusters (Fig. 3G-I, Supplementary Fig. S13, and Supplementary Table S2). We found that in this condition we found both the FeMoco intact and a similar density around the P-cluster as was found in the 60 mM dataset, indicating the P-cluster was fully reduced and in the $P^N$ state. Finally, most functional MoFeP assays use a concentration of 5 to 15 mM NaDT concentrations[56,57]. We then tested if 5 mM of NaDT could also protect MoFeP during freezing. We determined a final cryoEM MoFeP structure at 2.19 Å resolution and

observed that, as expected, the metal clusters were in their reduced states (Fig. 3J-L, Supplementary Fig. S14, and Supplementary Table S2). We speculate that lower concentrations of NaDT are capable of protecting the metal centers of MoFeP than those required to obtain deoxy Hb (e.g., 60 mM NaDT) is a result of the metal clusters of MoFeP being buried within the core of the protein, whereas the hemes in Hb are solvent-exposed. In addition, the time scales of oxygen damage to MoFeP and oxygen binding by Hb differ significantly. Hb promptly

binds O$_2$ in solution (within milliseconds), whereas the oxygen-induced oxidation of MoFeP occurs over longer timescales (tens of seconds). The presence of this intact P-cluster paired with the high resolution of this structure indicates that this method is well-suited for reproducibly preparing functionally anaerobic cryoEM grids and obtaining high-resolution structures of O$_2$-sensitive proteins.

## Discussion

The primary challenge in anaerobic protein preparation and cryoEM grid preparation lies in conducting both processes within an anaerobic environment (e.g., a glovebox/bag) or protecting an O$_2$-sensitive sample outside such an environment. Since most anaerobic purifications require careful handling within an anoxic environment, nearly all anaerobic cryoEM studies to date rely on relocating costly grid preparation devices into an anaerobic glovebox/bag. This approach creates several challenges. First, the temperature and humidity within an anaerobic chamber can be difficult to regulate. In addition, the transport of materials into a glovebox/bag can be challenging and time-consuming due to the requirement that each item that enters a glovebox/bag needs to undergo degassing cycles before it can enter the anaerobic environment. Here, we circumvent these problems by developing an (an)aerobic sample freezing protocol outside an anaerobic chamber using the SPT Labtech chameleon. We demonstrate the utility of this protocol by determining the high-resolution cryoEM structures of human deoxyHb and reduced *A. vinelandii* MoFeP, with both structures showing a lack of O$_2$ exposure during structure determination. During Hb preparation, we found that we could control the O$_2$ exposure in the samples by varying the concentration of NaDT in the cryoEM buffer. We observed that at NaDT concentrations below 5 mM, all four hemes within Hb were bound to O$_2$, while increasing [NaDT] to 20 mM or 60 mM led to either a partially-bound oxyHb structure or a fully deoxygenated Hb structure, respectively, representing a structure of partially O$_2$-bound Hb structure with $\alpha_1$ and $\beta_1$ hemes occupied. Similarly, for MoFeP, the absence of the protective oil layer and NaDT in the cryoEM buffer yielded oxidized P-clusters, but the addition of the protective oil layer and NaDT in the cryoEM buffer yielded reduced P-clusters. It should, however, be noted that with the inclusion of an O$_2$ scavenger, like NaDT, this protocol is optimized for the characterization of the reduced forms of the proteins of interest. While the oxidized states may be biologically relevant for certain O$_2$-sensitive proteins, accessing them with this protocol may be limited due to the reducing strength of NaDT. Although we have demonstrated this protocol to be effective for the chameleon, many aspects, including the layering of protective oil, are not currently amenable to traditional blot-and-plunge freezing protocols.

During the optimization of our anaerobic chameleon protocol, we found a few key steps within the chameleon operation that limit issues encountered during freezing. First, while the chameleon requires at least 5 μL of sample for operation, we recommend that the minimum volume of sample placed within the sample cup be 15 μl with 7 μL of that 15 μl being drawn up into the dispenser since the chameleon aspirates sample from the bottom of the sample cup. The additional volume of sample ensures that any O$_2$ contamination remains at the top of the sample vial near the protective oil layer and never enters the dispenser tip. Another issue that we overcame was the protective oil layer remaining on the dispenser after sample aspiration. Any oil remaining on the outside of the dispenser can affect sample deposition and, therefore, needs to be removed. During regular chameleon operation, the user is prompted to test the dispenser once the sample is aspirated. During this step, the user can re-prime the dispenser, increase sample application amplitude, and/or choose to clean and purge the system to remove debris. To ensure proper sample ejection after aspirating the sample through the protective oil layer, we found that we needed to perform multiple cleaning steps and adjust the amplitude to achieve the desired dispensing result. In addition, we

observed that the faster wicking time of 300 ms with adequate glow discharge was a good balance for obtaining sufficiently thin ice for high-resolution imaging. Furthermore, to limit the time the sample was exposed to O$_2$, we prioritized freezing grids within 90–120 s of aspiration and typically froze 1-2 grids per sample. Other than these changes, this protocol follows the same steps as an aerobic chameleon protocol and only requires 30–60 s additional operation time per grid prepared.

The SPT Labtech chameleon is an emerging instrument that is being adopted around the globe in over 20 cryoEM facilities, including the NIH-sponsored cryoEM national centers within the United States (Supplementary Table S3). We demonstrated how to adapt the chameleon workflow to keep samples (an)aerobic while generating high-quality grids amenable for high-resolution structure determination. Two primary modifications to the workflow – the protective oil layer and the addition of an O$_2$ scavenger – render this instrument amenable to the freezing of anaerobic samples. By having the ability to simply layer a sample with oil – which does not affect the quality of the prepared cryoEM grids – and supplement with a versatile chemical reductant, NaDT will allow any user studying O$_2$-sensitive proteins to create high-quality cryoEM grids using this (an)aerobic blot-free vitrification protocol.

## Methods

### Preparation of human metHb for cryoEM analysis
Lyophilized human methemoglobin (Sigma Aldrich) (metHb) was resuspended in phosphate-buffered saline (PBS; ThermoFisher) at pH 7.5 to a concentration of ~10 mg/mL. The concentration and oxidation state were measured and confirmed, respectively, using an Agilent 8453 UV-Vis spectrophotometer.

### Preparation of human deoxyHb for cryoEM analysis
MetHb was transferred into a Coy Laboratories anaerobic chamber (95% Ar, 5% H$_2$) and briefly degassed. A stock solution of 1 M sodium dithionite (NaDT) was prepared in 1 M Tris-base. NaDT was added to metHb to a final concentration of 100 mM and allowed to incubate for 5–10 min in the glove bag. DeoxyHb was then desalted into anaerobic PBS buffer in the glove bag using a PD-10 desalting column. The concentration and oxidation state were measured and confirmed, respectively, using UV-Vis. DeoxyHb was then diluted with anaerobic PBS to a final concentration of 125 μM Hb supplemented with either 25 μM, 20 mM, or 60 mM NaDT.

### Expression and purification of molybdenum iron protein (MoFeP) from *Azotobacter vinelandii*
Preparation of wild-type molybdenum iron protein (MoFeP) from *Azotobacter vinelandii* (*Av*MoFeP) (*Av* strain DJ200) was performed as previously detailed, and flash frozen under liquid nitrogen until use[58].

### CryoEM sample preparation
For Hb samples: 15 μL of human metHb (125 μM) was pipetted into a chameleon sample cup (SPT Labtech) and used for cryoEM grid preparation. 15 μL of human metHb (125 μM) was pipetted into a chameleon sample cup, and ~70 μL of Al's oil (Hampton Research) was carefully layered on top of the sample up to the brim of the sample cup and used for cryoEM grid preparation. In a Coy Laboratories anaerobic chamber (95% Ar, 5% H$_2$), 15 μL each of deoxHb with 25 μM [NaDT], deoxyHb with 5 mM [NaDT], deoxyHb with 20 mM [NaDT], and deoxyHb with 60 mM [NaDT] were separately pipetted into their own chameleon sample cup located within a 3 mL glass, conical-bottom Reacti-Vial (Thermo Fisher Scientific). The chameleon sample cups had equilibrated in the anaerobic chamber overnight. ~70 μL of anaerobic Al's oil that had equilibrated in the anaerobic chamber 72 + hrs was then carefully layered on top of each deoxyHb sample up to the brim of the sample cup. Each Reacti-Vial

sample was then sealed with a septum for transport to the chameleon for cryoEM grid preparation.

MoFeP that had been flash frozen in liquid nitrogen was transferred into a Coy Laboratories anaerobic chamber and exchanged into reaction buffer (20 mM Tris, pH 8.0, 25 mM NaCl, anaerobic). MoFeP was concentrated with a 100 kDa Microcon centrifugal filter (Millipore) to ~3 mg/mL and syringed-filtered (0.2 μM filter). Protein concentration was verified using an iron chelation assay where the protein was mixed with (6.2 M guanidine-HCl, 2 mM 2,2'-bipyridine, 10% glacial acetic acid) and absorption was measured at 522 nm. A final concentration was calculated using an extinction coefficient of 8650 M$^{-1}$ cm$^{-1}$ and assuming 30 Fe atoms per MoFeP. The protein was then diluted to 30 μM.

Stock detergent solutions of CHAPSO, fluorinated octylmaltoside (FOM), amphipol A8-35, and Brij-35 were prepared in separate vials and degassed in an anaerobic chamber. A detergent cocktail solution comprising 0.101% (w/v) CHAPSO, 0.00964% (w/v) FOM, 0.12% (w/v) A8-35, 0.06% (v/v) Brij-35, and either 0 or 40 mM NaDT was prepared[59]. 10 μL of the detergent cocktail mix was then carefully mixed with 10 μL of 30 μM MoFeP (1:1 final ratio) and 15 μL of the mixture was located into a chameleon sample cup within a 3 mL glass Reacti-Vial. The MoFeP sample without dithionite was then sealed within the Reacti-Vial using a septum for transport to the chameleon for cryoEM grid preparation. ~70 μL of anaerobic Al's oil was then carefully layered above the MoFeP sample containing either 5, 20, or 60 mM NaDT up to the brim of the sample cup, and the Reacti-Vial was sealed for transport to the chameleon for cryoEM grid preparation.

All samples were prepared and immediately taken to the chameleon for grid preparation. All steps were performed using a SPT Labtech chameleon. The chameleon was prepared according to established protocols. Liquid ethane was maintained within a temperature range of −173 to −175 °C, and the humidity shroud was maintained at >75% relative humidity (RH) during grid vitrification. For each sample, 2-4 chameleon self-wicking grids (Quantifoil Active 300 mesh; Quantifoil) were loaded into the instrument and glow discharged for 40–45 seconds at 12 mA. Once the instrument glow discharged the grids, the Reacti-Vial was unsealed, and the chameleon sample cup was immediately and carefully loaded into the instrument. 7 μL of the sample was immediately aspirated into the dispenser and subsequently tested for proper dispensing within the chameleon software before sample deposition onto the grid. At this stage, some Al's oil may remain on the dispenser tip blocking sample spray/deposition. Obvious signs of oil included oil-like features when "inspect dispenser" was selected, as well as a lack of sample droplets when reviewing dispenser testing in the software (Supplementary Fig. S1). If that was the case, then initially the dispense amplitude (under Advanced Troubleshooting) was increased 2-fold (~800) to attempt to remove the oil from the dispenser tip. The amplitude was consistently increased by 100 until sample droplets appear, or an amplitude of 1200 was reached. If droplets were observed, the amplitude can be decreased back down to proper working conditions (~400) to obtain the recommended 5 droplets. If, however, the oil persisted even with the increase in sample dispense amplitude to 1200, then the "Mini Prime" option was selected, followed by "Wipe Dispenser." At this point, the line should have been fully purged and relieved of oil. The amplitude of the dispenser was then adjusted until the recommended 5 droplets were obtained. Once ideal sample dispensing conditions were obtained, grids were frozen with 300–900 ms plunge time using 40–150 s glow discharge times. Of note, shorter wicking times required longer glow discharge times for proper ice thinning. The time from sample loading to vitrification of the first grid was 90–120 s. Each subsequent grid was frozen within 60–120 s of the first. We prioritized the first grid during screening and collection.

## Hb O$_2$ diffusion assay

For assessing O$_2$ diffusion into our samples, deoxyHb was prepared as described above and diluted with anaerobic PBS and NaDT to generate 1 mL each of the following samples: 1.5 μM deoxyHb with 25 μM NaDT, 1.5 μM deoxyHb with 20 mM NaDT; and 1.5 μM deoxyHb with 60 mM NaDT. The mixed samples were then transferred to a degassed plastic spectrophotometry cuvette and either measured as is or layered with anaerobic Al's oil that approximates the height of the oil layer added to the sample cup, approximately 500 μL. These samples were sealed and then transported out of the anaerobic environment and immediately placed into an Agilent 8453 UV-Vis spectrometer blanked with 1x PBS. For each sample, the cuvette was unsealed absorbance from 350 nm to 700 nm was measured with a time scan rate of 12000 nm/min at a wavelength interval of 2.5 nm for 20 min. The change in the absorbance of the Soret peak maximum at 415 nm was plotted vs time.

## CryoEM data collection

All data was acquired at UCSD's CryoEM Facility using a Titan Krios G4 (ThermoFisher Scientific) operating at 300 kV equipped with a Selectris-X energy filter (ThermoFisher Scientific). All images were collected at a nominal magnification of 165,000x in EF-TEM mode (with a calibrated pixel size of 0.735 Å) on a Falcon4 direct electron detector (ThermoFisher Scientific) using a 10 eV slit width. Micrographs were acquired in EER format with a cumulative electron exposure of 60 electrons/Å$^2$ with a defocus range of −1 to −2.0 μm. Data was collected automatically using EPU2 (ThermoFisher Scientific) with aberration-free image shift. Data was monitored live using cryoSPARC Live (Structura Bio), where movies were patch motion corrected and patch CTF estimated on the fly. Micrographs with a CTF estimation worse than 7 Å and/or a cumulative motion of more than 150 pixels were discarded. *Hb datasets:* the oiled met Hb dataset contained 2384 micrographs, while the non-oiled contained 2847 micrographs. The very low NaDT and low NaDT oxy Hb datasets contained 2475 and 1538 micrographs, respectively, the partial oxy Hb dataset contained 3925 micrographs, and the deoxyHb dataset contained 2220 micrographs. *MoFeP datasets:* the oxidized MoFeP dataset was two imaging sessions where dataset 1 contained 1067 micrographs and dataset 2 contained 1975 micrographs. The reduced 5 mM NaDT MoFeP dataset contained 2501 micrographs, and the 20 and 60 mM NaDT MoFeP datasets both contained 2000 micrographs.

## Processing

For all data sets (Hb and MoFeP), initial particle picking was performed within cryoSPARC Live using templates. These picks were then exported to cryoSPARC, and downstream processing for both Hb and MofeP proceeded as described below[60].

**Hb data processing.** For all Hb data sets (Oiled, Non-Oiled, 25 μM NaDT, 5 mM NaDT, 20 mM NaDT, 60 mM NaDT), particles were extracted at a box size of 256 pixels and Fourier cropped to 64 pixels at 2.94 Å/pixel. These particles were then subjected to one round of two-dimensional (2-D) classification, where obvious Hb tetramer classes were chosen to move forward. The selected particles we re-extracted and recentered using the same box and Fourier cropping and subjected to a second round of 2-D classification. The final set of selected particles from the 2-D classifications were then subjected to a non-uniform (NU) refinement using C$_2$ symmetry and a starting model from EMD-0407 (metHb) lowpass filtered to 15 Å and clipped to the same box size[40]. The particle stack was then subjected to heterogeneous refinement, with one volume being the volume from the previous NU refinement, the EMD-0407 volume, and the volume from EMD-4877 (20S proteasome)[40,61]. The particles that contributed to the best volume that resembled Hb tetramer were selected and extracted at a box size of 256 pixels with a Fourier crop to 128 pixels at 1.47 Å/pixel.

These re-extracted particles were subjected to an NU-refinement ($C_2$ symmetry) using the selected volume from the heterogeneous refinement low-pass filtered to 10 Å as an initial model to assess overall particle quality before fully unbinning the particles. Following this refinement, the particles were re-extracted with a box size of 256 pixels at 0.735 Å/pixel. These particles were then used for an ab initio model generation (2-classes) with a max resolution of 4 Å. The class with the best continuous density and completeness was chosen for a subsequent NU-refinement ($C_2$ symmetry) using the best ab initio volume low-pass filtered to 10 Å as an initial model. Additionally, this refinement optimized per particle defocus and aberrations. Finally, due to signal delocalization in real space that leads to loss of signal in real space as well as CTF aliasing in reciprocal space, we performed a final extraction at a box size of 352 pixels at 0.735 Å/pixel. We then performed an NU-refinement with $C_2$ symmetry and without (real space windowing off) with per particle defocus and aberration refinement turned on. These last two refinements lead to the final densities that were used for analysis and Phenix.resolve_cryo_em[49].

**MoFeP data processing.** For the oxidized MoFeP datasets, each micrograph set was template picked using templates generated from a previously published MoFeP volume[15] and extracted with a box size of 384 pixels Fourier cropped to 64 pixels at 4.41 Å/pixel. Dataset 1 particles were subjected to a 2-D classification. Particles in the best MoFeP classes were selected used for ab initio (2 classes) model generation. The ab initio class that provided the most complete MoFeP density was chosen, and the particles associated with this class were carried forward to a NU-refinement. We noticed that this resulting density had some streaking indicating the presence of destructive particles, hence we ran a second 2-D classification. We then performed another 2-class ab initio model generation where particles in the best class was used for a NU-refinement. The particles were re-centered and re-extracted with a box size of 384 pixels without any Fourier cropping resulting in a pixel size of 0.735 Å/pixel. These unbinned particles were subjected to an NU-refinement using the previous ab initio volume was as an initial model with per particle defocus and aberration refinement turned on (real space windowing off). The particles were then subjected to Reference Based Motion Correction (RBMC) and a final NU-refinement was performed using the same parameters as the previous NU-refinement before combining particles. For Dataset 2, we used a more simplified workflow that utilized a single round of 2-D classification followed by a 2-class ab initio model generation. Particles in the best class were then used for NU-refinement. Particles from Dataset 1 and Dataset 2 were then combined and re-extracted at a box size of 384 pixels at 0.735 Å/pixel. We then performed a NU-refinement on these combined particles, using the volume from the last NU-refinement of Dataset 1, with per particle defocus and aberration refinement turned on (real space windowing off). The combined particles were subjected to RBMC followed by a NU-refinement using the same parameters as the previous NU-refinement with $C_2$ symmetry and without. The densities from these NU-refinements were then used for analysis and Phenix.resolve_cryo_em.

Processing of the reduced MoFeP datasets was performed similarly as described for Dataset 2 of the oxidized MoFeP structure with the following exception: the "Import Beam Shift" tool was used to separate the particles into their different aberration free image shift groups (80 in total) for use during aberration refinement during NU-refinement.

**Refinement.** All models presented in this work were generated using a similar workflow. All final volumes were subjected to the phenix.resolve_cryo_em density modification tool, where only half maps and sequence file was supplied. For our metHb, oxyHb, mixed Hb, and deoxyHb maps, PDB: 6NBC was fit into the resulting RESOLVE density and manually adjusted in COOT. For the oxidized and reduced MoFeP

structures, PDB: 7UT7 was fit into the resulting RESOLVE density and manually adjusted in COOT $O_2$'s were manually placed in the OxyHb and mixed Hb structures. Realspace refinement was performed using Phenix.[62] Parameter files for the heme-$O_2$, P-cluster ($P^N$ or $P^{2+}$), and homocitrate-FeMoco ligands were used during refinement. Phenix douse was then used on the resulting model to identify and add water. The model was then checked for accuracy in COOT, and a final real space refinement was performed against the sharpened map from cryoSPARC (Supplementary Table S1).

## Reporting summary
Further information on research design is available in the Nature Portfolio Reporting Summary linked to this article.

## Data availability
The data that support this study are available from the corresponding authors upon request. Structural models have been deposited in the Protein Data Bank (PDB) with accession codes: 9CQO(non-oiled metHb, $C^1$), 9CQP(non-oiled metHb, $C_2$), 9CQQ (oiled metHb, $C_1$), 9CQR(oiled metHb, $C_2$), 9CQM(very low (25 μM) NaDT Hb, $C_1$), 9CQN(very low (25 μM) NaDT Hb, $C_2$), 9CQS(low (5 mM) NaDT Hb, $C_1$), 9CQT(low (5 mM) NaDT Hb, $C_2$), 9CQU(medium (20 mM) NaDT Hb, $C_1$), 9CQV(high (60 mM) NaDT Hb, $C_1$), 9CQW(high (60 mM) NaDT Hb, $C_2$), 9CQX(oxidized MoFeP, $C_1$), 9CQY(oxidized MoFeP, $C_2$), 9MLY(reduced (5 mM NaDT) MoFeP, $C_1$), 9MLZ(reduced (5 mM NaDT) MoFeP, $C_2$), 9CQZ(reduced (20 mM NaDT) MoFeP, $C_1$), 9CR0(reduced (20 mM NaDT) MoFeP, $C_2$) and 9MM0(reduced (60 mM NaDT) MoFeP, $C_1$), 9MM1(reduced (60 mM NaDT) MoFeP, $C_2$). PDB models for comparing heme densities have accession codes: 2DN1(oxy human Hb) and 2DN2(deoxy human Hb). The corresponding cryoEM maps are available at the Electron Microscopy Data Bank (EMDB) with accession codes EMD-45817(non-oiled metHb, $C_1$), EMD-45818(non-oiled metHb, $C_2$), EMD-45819(oiled metHb, $C_1$), EMD-45820(oiled metHb, $C_2$), EMD-45815(very low (25 μM) NaDT Hb, $C_1$), EMD-45816(very low (25 μM) NaDT Hb, $C_2$), EMD-45821(low (5 mM) NaDT Hb, $C_1$), EMD-45822(low (5 mM) NaDT Hb, $C_2$), EMD-45823(medium (20 mM) NaDT Hb, $C_1$), EMD-45824(high (60 mM) NaDT Hb, $C_1$), EMD-45825(high (60 mM) NaDT Hb, $C_2$), EMD-45826(oxidized MoFeP, $C_1$), EMD-45827(oxidized MoFeP, $C_2$), EMD-48381(reduced (5 mM NaDT) MoFeP, $C_1$), EMD-48382(reduced (5 mM NaDT) MoFeP, $C_2$), EMD-45828(reduced (20 mM NaDT) MoFeP, $C_1$), EMD-45829(reduced (20 mM NaDT) MoFeP, $C_2$), and EMD-48383(reduced (60 mM NaDT) MoFeP, $C_1$), EMD-48384(reduced (60 mM NaDT) MoFeP, $C_2$). EM densities for comparing heme densities are available at EMD-37575(oxy human Hb) and EMD-37576(deoxy human Hb). All other data are available in the main text or the Supplementary Materials.

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

## Acknowledgements

We are grateful to Prof. Kevin Corbett for providing valuable feedback on this manuscript, and the entirety of the Herzik lab for facilitating insightful discussions. We thank Marc Morizono for his artistic contribution to Fig. 1A. The authors acknowledge the facilities, along with the scientific and technical assistance of the staff of the cryo-EM facility at UC San Diego, Dr. Mariusz Matyszewski and Dr. Inga Kuschnerus. We also thank Brendan Dennis, Kevin Smith, and the UCSD Physics Computing Facility for their insights and support. Molecular graphics and analyses were performed with UCSF ChimeraX, developed by the Resource for Biocomputing, Visualization, and Informatics at the University of California, San Francisco (UCSF), with support from National Institutes of Health (NIH) grant R01-GM129325 and the Office of Cyber Infrastructure and Computational Biology, National Institute of Allergy and Infectious Diseases. This work was supported from the NIH grant R35-GM138206 (MAH), NIH grant R01-GM148607 (FAT and MAH). The chameleon used in this protocol was obtained via NIH 1S10OD032471 (MAH). SMN is supported by the NIH Interfaces Graduate Training Program at UCSD (T32-EB009380). BC is supported by the Goeddel Family Technology Sandbox Fellowship.

## Author contributions

B.D.C., S.M.N., and K.L.M. conceived the project, designed experiments, performed protein isolation/characterization, cryoEM experiments analysis, and co-wrote the manuscript. Y.L. performed protein isolation/characterization and Hb O$_2$ diffusion assay and analysis. F.A.T. provided experimental guidance, data analysis, and edited the manuscript. M.A.H. conceived and directed the project, designed experiments, oversaw cryoEM experiments, performed data analysis, and co-wrote the manuscript. All authors participated in the editing of the manuscript.

## Competing interests

The authors declare no competing interests.
