## [Transparent Peer Review file · Nature Communications]

Preparation of oxygen-sensitive proteins for high-resolution cryoEM structure determination using blot-free vitrification

Corresponding Author: Dr Mark Herzik, Jr

Version 0:

Reviewer comments:

Reviewer #1

(Remarks to the Author)

Manuscript NCOMMS-24-51012-T

In this manuscript, Cook, Narehood and collaborators present an original and promising method for the preparation of single-particle cryogenic electron microscopy (cryoEM) grids under anaerobic conditions, specifically for the study of oxygen-sensitive metalloproteins. Instead of using grid preparation systems that require blotting time and are integrated directly into an anaerobic chamber, they propose a method that can be applied to any blot-free system, such as Chameleon or Spotiton, likely available on cryoEM platforms, without the need for an anaerobic chamber, except during the initial sample preparation step. To prevent damage caused by oxygen exposure, they suggest covering the sample with a mixture of oils to prevent oxygen diffusion during the pipetting step. The critical point of oxygen exposure occurs during the deposition and auto-wicking of the sample, where the surface-to-volume ratio is highest, making it most susceptible to oxygen contamination. To validate their method, the authors use two control samples: human hemoglobin and nitrogenase MoFe-protein.

The experimental work was carried out rigorously, and the structures determined by cryoEM are of very high quality. The manuscript clearly presents the obtained results, and the methods are described in great detail. However, the conclusion of this work reveals that it is essential to add at least 20 mM of sodium dithionite (Na-DT) to the sample to eliminate any contamination during the auto-wicking step, which represents a significant limitation to the described approach, particularly regarding the goal of generalizing the method for studying the reaction mechanisms of these metalloenzymes. In fact, 20-60 mM Na-DT is 4-10 times the concentration commonly used by various research groups during the preparation (purification) of their samples, which is quite substantial. Moreover, Na-DT is a strong reducing agent, and as demonstrated by the authors, under these conditions, the obtained structures correspond to the so-called reduced states of hemoglobin or nitrogenase. Thus, while the method has the significant advantage of being able to protect most samples from oxygen damage and thereby allow their structural study by cryoEM, the method proposed by the authors does not preserve the oxidized states of these metalloproteins, which will therefore be excluded from possible analysis.

The reviewer acknowledges that this new method offers the possibility for various research groups to achieve structural characterization of their metalloproteins at a lower cost. However, the generalization of this approach as a solution for detailed studies of metalloprotein mechanisms is not feasible, as only the reduced states are accessible. For certain non-redox metalloproteins, the reduction of metal centers may be detrimental or even deleterious. The authors do not address this aspect at all in their manuscript. It is clear that this work deserves publication for the benefit of the community, but it is important that the authors clarify that only reduced states will be accessible, or they should provide a solution to include the study of oxidized states. If these modifications are made, the reviewer supports the publication of this manuscript in Nature Communications.

Reviewer #2

(Remarks to the Author)

Cook et al., describes the development of single particle cryoEM grid preparation methodology for oxygen-sensitive proteins using a commercial vitrification device, the SPT Labtech chameleon in an aerobic environment, termed (an)aerobic

vitrification. This may be of interest to researchers working on structural characterization of air-sensitive biochemical systems. The authors use the oxygen-reactive human hemoglobin (Hb) and the nitrogenase MoFe protein (MoFeP) as test cases. The authors report that using the chameleon under normal conditions results in oxidized states for both Hb and MoFeP, but by using an anaerobic chamber for sample preparation, addition of an oil layer and excess of the reductant sodium dithionite, and optimization of vitrification settings, they are able to achieve reduced states for both proteins. While the authors successfully demonstrate that they are able to achieve excellent, high resolution cryoEM structures of both Hb and MoFeP in this system, it remains to be determined whether this approach constitutes a significant advancement over the perhaps less automated but nonetheless air tight methods previously reported by Cherrier et al., 2022 (Biomolecules), and Warmack et al., 2024 (Nat. Protoc.), as well as in a recent paper by Schmidt et al., 2024 (NSMB). In contrast to methods that leverage routine freezing instruments placed in anaerobic tents, this approach requires costly equipment including an SPT Labtech chameleon, which costs ~4x more than a Vitrobot (as used in the aforementioned protocols), and the approach still requires anaerobic chambers. Additionally, this protocol appears to require a high excess of dithionite to preserve reduced states of the protein. The latter point is of concern for those interested in exploring the catalytic turnover conditions mentioned by the authors, as this reductant is notorious for decomposition in aqueous solution to a variety of breakdown products which may affect interpretation of these experiments (recently reported for nitrogenase by Bilyj et al., 2024). It is also not entirely clear that the Hb structure is completely deoxygenated in the optimized conditions. Overall the authors show that specific conditions can be result in reduced, or partially reduced cryoEM structures using a Chameleon, but the general applicability of the method remains in question. The two structures determined in this effort serve as well-known and important controls, the application of the approach to novel structures will be its true test.

Additional comments:

1. Within the series of Hb structures, the T-state (deoxygenated) is never achieved, even while there is no oxygen modeled into the final 60 mM DT/Hb structure. Why is this state never reached if this represents a fully deoxygenated structure? Ideally there would be control structure that is prepared fully anaerobically to see if the T state can be determined.
2. The work highlights a series of Hb cryoEM structures, which are excellent quality for such a small protein by cryoEM. However, even at these resolutions (2.4 – 2.7 Å), some observations may need secondary confirmation. Specifically, the authors suggest that in the 60 mM DT/Hb condition there is no oxygen bound, but that there is density for a nearby (and overlapping) water molecule, which is shown in supplemental data figure 9 (Aside: why is the density for the water not shown in figure 2?). However, at 2.7 Å resolution it is not clear that whether this is mixed occupancy water/oxygen. This density also looks very similar to that shown in subunit beta1 of the 20 mM condition, which is modeled as an O₂. Calculating difference maps between the map and model with either O₂ or water modeled may provide some additional support that the density seen in the 60 mM DT map is indeed a water, but may still not be definitive.
3. In Figure 2, the 20 mM DT/Hb structure has C1 applied and shows varying oxygen occupancy between the two halves of the protein. The C1 density should be shown for each of these Hb structures and the occupancy assessed between all halves.
4. While the authors are able to achieve high resolution (sub 3 Å) for all their tests, this may not be achievable for all samples, nor do all oxygen-sensitive proteins demonstrate a notable conformational shift upon oxidation. For researchers achieving lower resolution or looking at different proteins, a preliminary experiment with an oxygen-sensitive dye to vet the system may be more applicable.
5. Related to point 4, as oxidation may have to proceed to a significant degree before it becomes visible within the Hb and MoFeP structures, it may be informative to perform a series of activity measurements while the protein is kept under oil to see whether or not the proteins are losing activity even if they may not be oxidized enough to show a conformational change.
6. The author's note in the discussion that they freeze 1-2 grids per sample, and that the protocol requires a higher volume of sample than is typical for the chameleon. This may be limiting for researchers with lower yield, it would be good to comment on this.
7. Why is 20 mM dithionite concentration used for the nitrogenase protein, while 60 mM dithionite was found to be necessary for Hb? Comment on this.
8. A minor point, in the Figure 2 legend the resolution of the 60 mM DT/Hb map is listed as 2.7 Å, while in the text it is listed as 2.6 Å, and in the figure legend it is not stipulated whether or not the map shown is C1 or C2. These points should be clarified.
9. Within the abstract the authors state: "Current practices for anaerobic grid preparation involve a vitrification device located in an anoxic chamber, which presents significant challenges including temperature and humidity control" – however, the anaerobic cryoEM protocols published to date (Cherrier et al., Warmack et al., Schmidt et al., 2024) utilized vitrobots within anaerobic chambers, which control temperature and humidity. This point should be reconsidered or removed from the abstract.
10. There appears to be a type at the end of the Figure 2 legend, “\”
11. Line 294: 'alpha' should be the symbol for alpha.

12. Explain the coloring scheme used for the alpha and beta subunits in Figure 2.

Reviewer #3

(Remarks to the Author)

In the present manuscript, Cook et. al. test the suitability of a blot-free vitrification device, the SPT Labtech chameleon (hereafter referred to as chameleon), for the preparation of oxygen (O₂) sensitive cryogenic electron microscopy (cryoEM) samples. Using hemoglobin as an O₂ reporter and the catalytic component of the molybdenum nitrogenase (MoFe protein) as an O₂ sensitive proof of principle sample. The authors outline a very useful workflow for the preparation of O₂ sensitive cryoEM samples with the chameleon in a regular aerobic atmosphere with a protective oil layer and in the presence of sodium dithionite, thus enabling a wider range of scientists to freeze O₂ sensitive cryoEM samples.

As discussed, anaerobic preparation of cryoEM samples by today's standards is not optimal. To prevent O₂-induced damage to the sample, a vitrification device such as a Vitrobot is placed inside an anoxic chamber, which allows for sample preparation under an anoxic atmosphere but causes problems in humidity/temperature control, handling of the samples, etc. However, despite these drawbacks the preparation of cryoEM samples under an anoxic atmosphere ensures the anaerobic sample preparation, whereas the approach presented in this paper might not. The chameleon is operating under an aerobic atmosphere. Therefore, preparation of hemoglobin and MoFe protein samples without any precautions yielded O₂ bound hemoglobin and O₂ damaged MoFe protein, respectively. The authors introduce two measures to overcome this. First, they place an Al's oil layer on top of the sample reservoir that prevents O₂ diffusion into the sample. Second, they add the reductant sodium dithionite (NaDT) to their samples, a well-known O₂ scavenger that is widely used to protect O₂ sensitive samples such as the nitrogenase enzyme. In my opinion, these measures are not sufficient to refer to the overall process as anaerobic, also not as (an)aerobic. The moment the sample is aspirated by the chameleon, it is exposed to an aerobic environment. As demonstrated by the authors, the added NaDT is scavenging most of the O₂ during the sample preparation process, thus high amounts NaDT are required to keep the samples reduced (60mM and 20mM for hemoglobin and MoFe protein samples, respectively). Hence, I would rather refer to the overall process as aerobic in the presence of high amounts of an O₂ scavenger.

Overall, the manuscript is well written and the authors present their data in a convincing manner. However, several concerns need to be addressed prior to publication:

1. My main concern regards the expression "(an)aerobic blot free vitrification". The word "(an)aerobic" is problematic as discussed above and should be deleted in the title and the abstract. However, "(an)aerobic" could potentially be used throughout the text after the explanations in line 73f. Moreover, I am wondering, whether it would be possible to conduct the whole process under an anoxic atmosphere? Judging by the demonstration video of the device that is available online, the vitrification process does operate in a chamber. Could one maybe modify the device setup in a way that the chamber is constantly flushed with argon or nitrogen? Making the whole process really anaerobic and with this offering a brilliant solution for O₂ sensitive cryoEM samples.
2. The authors state in the abstract (line 27), the introduction (line 68) and later in the main text that the conventional way of preparing cryoEM grids anaerobically (Vitrobot inside an anaerobic chamber) is costly and not accessible to the community. In line with this argumentation, the authors present their new workflow as a more cost-effective solution and argue that the chameleon is already widely spread across cryoEM facilities and thus more accessible. However, I am missing a direct cost comparison of the two approaches. What's the cost for purchasing and operating the chameleon? How does that compare to the conventional setup of a Vitrobot inside an anoxic chamber? Approximately how many chameleons are operating globally and can this potentially be compared to the number of Vitrobots or anoxic chambers? At least the price of the chameleon and vitrobot inside the tent should be compared, if so much emphasize is put on the cost advantage.
3. Fig. 3: I assume that the chameleon device is more expensive than a Vitrobot. Given that the chameleon is also operating under an aerobic atmosphere, I am wondering if the same results obtained with the chameleon could be obtained with a Vitrobot operating under aerobic conditions, using the same O₂ protection measures established for the chameleon (20mM NaDT and a protective oil layer). Is it possible to obtain high-resolution structures of the reduced MoFe protein under these conditions with a Vitrobot? This would circumvent all problems described for operating a Vitrobot inside an anoxic chamber. Moreover, Vitrobots are the standard vitrification device and thus the most accessible. Demonstrating that this workflow would also work for a Vitrobot would significantly increase the impact of this manuscript.
4. Line 92: The authors point out that the chameleon might be able to help with preferential orientation problems. It is known that the MoFe protein is prone to interact with the air-water interface during cryoEM sample preparation and is thus prone to preferential orientation issues (Warmack, R. A., & Rees, D. C. (2022). Anaerobic single particle cryoEM of nitrogenase. *bioRxiv*, 2022.2006.2004.494841. doi:10.1101/2022.06.04.494841). As described in the materials and methods (line 410) the authors used a detergent cocktail in their sample preparation for the MoFe protein structures presented in Fig. 3. Detergents are commonly used in cryoEM to alleviate the effects of preferential orientation, thus it is not clear whether the chameleon helped to overcome this challenge or the the detergent cocktail. To adress this question: Is it possible to get a high-resolution structure of the MoFe protein without the use of detergent when using the chameleon for sample preparation?

Minor comments:

- Please add the missing "S" to Table 1 and Table 2 when referencing Table S1 and S2.
- Line 192: "...predicted concentration of O₂ (250μM) in solutions...". I am missing a reference for this statement.

- Line 277: "... we decided to use MoFeP to determine if our anaerobic freezing protocol...". Brackets for (an)aerobic missing.
- Line 249: "Although the overall tetramer structure was once again R state, our analysis of the heme groups in each subunit revealed an absence of any density stretching above the Fe, indicative of O₂ binding within the hemes." Please rephrase to clarify, for example: "...any density stretching above the Fe, which would have been indicative of O₂ binding within the hemes."
- Line 340: "Any oil remaining on the outside of the dispenser can affect sample deposition and therefore needs to be removed". From the results described in line 138 to 163 I concluded that the oil does not adversely affect the cryoEM samples. Were "additional dispenser wash steps to remove any residual oil" (line 142) applied in these experiments? If yes, I would recommend to clarify that this is an essential step in the chameleon process.
- Line 405: Please describe the procedure for the iron chelation assay or provide a reference.
- Line 527: "- missing between "NU" and "refinement".

Reviewer #4

(Remarks to the Author)

Version 1:

Reviewer comments:

Reviewer #1

(Remarks to the Author)

In this revised version of their manuscript, the authors have addressed all my previous concerns. As highlighted in my earlier review, this manuscript provides a clear and detailed description of the methods employed, making it a valuable resource for the scientific community. By presenting an approach that enables structural characterization of metalloproteins at a reduced cost, this work has the potential to significantly benefit a wide range of research groups.

However, as the authors themselves acknowledge, the implementation of such a device is currently concentrated in facilities predominantly located in the United States. This geographical concentration may limit access to this method for researchers outside these regions due to the high cost of the equipment.

In addition, it is worth noting that integrating a grid vitrification device within an anaerobic chamber, while more complex and expensive, could offer exciting opportunities for studying specific redox states in metalloproteins. While probably confined to specialized research groups, this alternative approach expands the possibilities for structural characterization.

Having said this, I am convinced that this manuscript represents a substantial contribution to the field and I fully support its publication in Nature Communications.

Reviewer #2

(Remarks to the Author)

I thank the authors for their important contributions in making the community aware of these important resources that are valuable to explore for the growing field of cryoEM. However, for the benefit of the community, I urge the authors to be balanced and forthcoming in their discussion of the limitations of their approach.

As my fellow reviewers have noted, certain facts are indisputable about the work as presented:

(1) The process of using a chameleon for cryoEM sample preparation is not shown here to be truly anaerobic, as demonstrated by the oxidation of the Hb when lower concentrations of reductant are added. Therefore, I strongly encourage the authors to replace the term "(an)aerobic" in their title, as it is not germane to the community and is ill defined in this context.

(2) The availability of semi-automatic freezing robots (GP2 and Vitrobot) exceeds that of the newer and more advanced Chameleons by at least an order of magnitude. While they may not be in anaerobic chambers, the authors should test their methodology with this more canonical equipment. The samples could easily be overlaid with oil, and blotting times of less than a second can be achieved. As we see from the data, the speed of oxygen diffusion across the thin layer of liquid created by a printing robot would exceed the time delay before freezing, making the argument against the comparison to slower freezing methods moot.

It is true that a vitrobot in an anaerobic chamber is not ideally suited for routine use in a user facility, and perhaps some variation of the proposed protocol herein is better. However, that may require additional optimization to limit oxygen exposure, improve sample exchange, etc.

Reviewer #3

(Remarks to the Author)

Thank you for revising the manuscript and addressing my feedback and concerns. While I do believe the manuscript has improved significantly, I would like to highlight some points that were only insufficiently addressed. The respective points are

according to the numbering of the authors:

1. Regarding the term (an)aerobic used in the title and the abstract: I have no doubt that the conditions used in the study can prevent oxygen-induced damage to oxygen sensitive samples. Moreover, I fully agree with the authors that the conditions are “not truly anaerobic by definition”, therefore I am convinced that the term “(an)aerobic” should be used carefully. In fact, I find it misleading that such a new and undefined expression is used without explanation in the title and abstract of the paper. Thus, I request to rephrase the title and introduce the term “(an)aerobic” together with the explanation provided in line 75 in the abstract. For the title I suggest “Preparation of reduced oxygen-sensitive proteins for high-resolution cryoEM structure determination using blot-free vitrification under air”. I am convinced that this title would be more clear and more interesting for a general readership.

4. Thank you for providing a cost comparison of the presented method with the use of a Vitrobot in a tent in the rebuttal letter. However, I am missing this comparison in the discussion of the current manuscript version, as promised in the rebuttal letter. Please add the cost comparison to the manuscript. Moreover, please add to Table S3 also the location (country and city) not only the name of the institute.

7. As the authors confirmed in the answer to my comment, the detergent cocktail is essential for overcoming the air-water interface issues of nitrogenase samples. Thus, the authors should explicitly state that the detergent cocktail is essential for the nitrogenase samples in the manuscript, for example in the paragraph from lines 289 – 309.

L. 320, correct “determinized” to “determined”

Reviewer #4

(Remarks to the Author)

REVIEWER COMMENTS

Reviewer #1 (Remarks to the Author):

Manuscript NCOMMS-24-51012-T

In this manuscript, Cook, Narehood and collaborators present an original and promising method for the preparation of single-particle cryogenic electron microscopy (cryoEM) grids under anaerobic conditions, specifically for the study of oxygen-sensitive metalloproteins. Instead of using grid preparation systems that require blotting time and are integrated directly into an anaerobic chamber, they propose a method that can be applied to any blot-free system, such as Chameleon or Spotiton, likely available on cryoEM platforms, without the need for an anaerobic chamber, except during the initial sample preparation step. To prevent damage caused by oxygen exposure, they suggest covering the sample with a mixture of oils to prevent oxygen diffusion during the pipetting step. The critical point of oxygen exposure occurs during the deposition and auto-wicking of the sample, where the surface-to-volume ratio is highest, making it most susceptible to oxygen contamination. To validate their method, the authors use two control samples: human hemoglobin and nitrogenase MoFe-protein.

The experimental work was carried out rigorously, and the structures determined by cryoEM are of very high quality. The manuscript clearly presents the obtained results, and the methods are described in great detail. However, the conclusion of this work reveals that it is essential to add at least 20 mM of sodium dithionite (Na-DT) to the sample to eliminate any contamination during the auto-wicking step, which represents a significant limitation to the described approach, particularly regarding the goal of generalizing the method for studying the reaction mechanisms of these metalloenzymes. In fact, 20-60 mM Na-DT is 4-10 times the concentration commonly used by various research groups during the preparation (purification) of their samples, which is quite substantial.

1. We thank for the reviewer for their time and suggestions. We have updated our draft to show that we are able to determine the ~ 2.2 Å structure of reduced MoFeP using our protocol with 5 mM NaDT concentration (Figure 3D; EMDB-48382) that indicates the metal clusters remain reduced throughout the experiment. This concentration was used in previous nitrogenase studies, including when determining catalytic activity, and demonstrates the same protective effect as the 20 mM and 60 mM NaDT samples (Schmidt et al., 2024; Sippel and Einsle, 2017). We have updated the language in the manuscript to also indicate that users can tailor the concentration of NaDT based on their protein(s) of interest from as low as 5 mM to 60 mM.

Updated Figure 3.

Moreover, NaDT is a strong reducing agent, and as demonstrated by the authors, under these conditions, the obtained structures correspond to the so-called reduced states of hemoglobin or nitrogenase. Thus, while the method has the significant advantage of being able to protect most samples from oxygen damage and thereby allow their structural study by cryoEM, the method proposed by the authors does not preserve the oxidized states of these metalloproteins, which will therefore be excluded from possible analysis. The reviewer acknowledges that this new method offers the possibility for various research groups to achieve structural characterization of their metalloproteins at a lower cost. However, the generalization of this approach as a solution for detailed studies of metalloprotein mechanisms is not feasible, as only the reduced states are accessible. For certain non-redox metalloproteins, the reduction of metal centers may be detrimental or even deleterious. The authors do not address this aspect at all in their manuscript.

2. The reviewer has a valid point in that the inclusion of an oxygen scavenger, like NaDT, in this protocol is optimized for the characterization of the reduced form of our proteins of interest. This is of course contingent upon the fact that the oxygen scavenger used has the reducing power necessary for the reduction of the protein sample. In this case, NaDT is such a strong reducing agent (-400 mV at pH 7) we cannot avoid reduction of our proteins, and for the most part, the majority of metalloproteins (Bollella and Katz, 2020). We have adjusted our narrative to reflect that with this current protocol most protein samples are limited to the structural characterization of their corresponding reduced states.
3. We also would like to point out that oxidized forms of many metalloproteins can be obtained under aerobic conditions using current hardware (Katsyv et al., 2023; Shi et al., 2020; Suzuki et al., 2023).

It is clear that this work deserves publication for the benefit of the community, but it is important that the authors clarify that only reduced states will be accessible, or they should provide a solution to include the study of oxidized states. If these modifications are made, the reviewer supports the publication of this manuscript in Nature Communications.

4. We thank the reviewer for their time and thoughtful comments. We have updated the manuscript to reflect that our protocol allows for accessing oxygen-free reduced states of proteins.

Reviewer #2 (Remarks to the Author):

Cook et al., describes the development of single particle cryoEM grid preparation methodology for oxygen-sensitive proteins using a commercial vitrification device, the SPT Labtech chameleon in an aerobic environment, termed (an)aerobic vitrification. This may be of interest to researchers working on structural characterization of air-sensitive biochemical systems. The authors use the oxygen-reactive human hemoglobin (Hb) and the nitrogenase MoFe protein (MoFeP) as test cases. The authors report that using the chameleon under normal conditions results in oxidized states for both Hb and MoFeP, but by using an anaerobic chamber for sample preparation, addition of an oil layer and excess of the reductant sodium dithionite, and optimization of vitrification settings, they are able to achieve reduced states for both proteins.

While the authors successfully demonstrate that they are able to achieve excellent, high resolution cryoEM structures of both Hb and MoFeP in this system, it remains to be determined whether this approach constitutes a significant advancement over the perhaps less automated but nonetheless air tight methods previously reported by Cherrier et al., 2022 (Biomolecules), and Warmack et al., 2024 (Nat. Protoc.), as well as in a recent paper by Schmidt et al., 2024 (NSMB). In contrast to methods that leverage routine freezing instruments placed in anaerobic tents, this approach requires costly equipment including an SPT Labtech chameleon, which costs ~4x more than a Vitrobot (as used in the aforementioned protocols), and the approach still requires anaerobic chambers.

1. We thank the reviewer for their insights. We acknowledge the cost difference between the two instruments; however, we want to highlight that the protocols referenced by the reviewer require the sequestration of freezing devices into dedicated anaerobic chambers, typically in a space outside of a core facility and away from ordinary users. Our protocol would allow for the anaerobic user to easily integrate with aerobic users using instrumentation housed and operated within core cryoEM facilities. We have updated the text to reflect that there are over 20 chameleon devices across the world – including 15 instruments at national centers regional centers in the US – that are available to the broader cryoEM community for free use through proposal requests. We have added an additional table (Table S3) listing the location of these devices. The goal for this protocol is for a user to be able to prepare a sample in an anaerobic environment and then simply transport it to any chameleon device and be able to freeze high quality grids.

Additionally, this protocol appears to require a high excess of dithionite to preserve reduced states of the protein. The latter point is of concern for those interested in exploring the catalytic turnover conditions mentioned by the authors, as this reductant is notorious for decomposition in aqueous solution to a variety of breakdown products which may affect interpretation of these experiments (recently reported for nitrogenase by Bilyj et al., 2024).

2. As mentioned in our response to Reviewer 1 (comment #1.1), we have updated the manuscript to include a ~ 2.2 Å MoFeP structure obtained under a low NaDT concentration (5 mM) that shows nitrogenase remains reduced under these conditions. This is a typical NaDT concentration used in previous studies.

It is also not entirely clear that the Hb structure is completely deoxygenated in the optimized conditions.

3. We have updated Figure 2 to show the C1 maps obtained under each condition. We also included a supplemental figure (Figure S11) comparing the heme groups obtain under each condition to previously published cryoEM and X-ray crystallography structures. These figures present strong evidence that the hemes in the deoxy condition lack oxygen, while the density for oxygen in the oxygenated samples match previous findings.

Updated Figure 2.

Overall, the authors show that specific conditions can result in reduced, or partially reduced cryoEM structures using a Chameleon, but the general applicability of the method remains in question. The two structures determined in this effort serve as well-known and important controls, the application of the approach to novel structures will be its true test.

4. The two proteins chosen in this study are quite distinct in their properties – Hb binds oxygen but undergoes slow oxidation under aerobic conditions while MoFeP does not stably bind oxygen but undergoes rapid oxidation under aerobic conditions. We feel that these model proteins are distinct enough to justify broad applicability. Of course, each user will need to optimize certain aspects of the protocol (e.g., concentration and chameleon wicking settings) for their specific specimen of interest.

Additional comments:

1. Within the series of Hb structures, the T-state (deoxygenated) is never achieved, even while there is no oxygen modeled into the final 60 mM DT/Hb structure. Why is this state never reached if this represents a fully deoxygenated structure? Ideally there would be control structure that is prepared fully anaerobically to see if the T state can be determined.

5. We thank the reviewer for their close analysis of our structures. While in the literature, T-state is thought to be the resting state of deoxygenated state of hemoglobin, this state can only be stably achieved using allosteric regulators (see commentary in Takahashi et al. 2024). Additionally, there are structures of deoxygenated R-state hemoglobin structures determined using X-ray crystallography that are similar to our structure presented herein (Paoli et al., 1996; Wilson et al., 1996). These structures show that the presence of either state does not exclude oxygen binding. Additionally, it has been show T-state Hb has a lower affinity for oxygen in solution than R-state (Bettati and Mozzarelli, 1997). Since our study is focused on oxygen contamination during the freezing process, using a higher affinity oxygen binder is beneficial for testing conditions. To ensure that the Hb maps and models determined in this study accord with previous literature, we have included a new supplemental figure (Figure S10) where we have isolated each heme from the maps in this study and have included both a recent cryoEM study (Takahashi et al., 2024) on human Hb that has both oxy and deoxy structures, as well as high resolution (1.25Å) crystal structures (Park et al., 2006). Comparing the densities above the hemes in this study to the published map reveals no significant differences. Finally, in correspondence with the authors of the recent cryoEM Hb paper ((Takahashi et al., 2024), the use of lyophilized Hb prevents the formation of T-state even in the presence of allosteric regulators. However, as demonstrated in this study, lyophilized hemoglobin retains its ability to bind oxygen and can be resolved at high resolution We have updated the text to reflect these findings.

2. The work highlights a series of Hb cryoEM structures, which are excellent quality for such a small protein by cryoEM. However, even at these resolutions (2.4 – 2.7 Å), some observations may need secondary confirmation. Specifically, the authors suggest that in the 60 mM DT/Hb condition there is no oxygen bound, but that there is density for a nearby (and overlapping) water molecule, which is shown in supplemental data figure 9 (Aside: why is the density for the water not shown in figure 2?). However, at 2.7 Å resolution it is not clear that whether this is mixed occupancy water/oxygen. This density also looks very similar to that shown in subunit beta1 of the 20 mM condition, which is modeled as an O2. Calculating difference maps between the map and model with either O2 or water modeled may provide some additional support that the density seen in the 60 mM DT map is indeed a water, but may still not be definitive.

6. We have updated Figure 2 to accurately represent the findings from each condition. We incorrectly assigned an O₂ in the 20 mM condition, and this has now been changed to a water. Our additional supplementary figure (Figure S10) allows for close examination of each heme and allows us to be confident in the presence or absence of O₂.
- 3. In Figure 2, the 20 mM DT/Hb structure has C1 applied and shows varying oxygen occupancy between the two halves of the protein. The C1 density should be shown for each of these Hb structures and the occupancy assessed between all halves.**
7. We have updated our Figure 2 to now include all C1 maps and models. (Please see comment #2.3)
- 4. While the authors are able to achieve high resolution (sub 3 Å) for all their tests, this may not be achievable for all samples, nor do all oxygen-sensitive proteins demonstrate a notable conformational shift upon oxidation. For researchers achieving lower resolution or looking at different proteins, a preliminary experiment with an oxygen-sensitive dye to vet the system may be more applicable.**
8. We appreciate the reviewer's experimental suggestion. We attempted to use an oxygen-sensitive dye to assess oxygen contamination during freezing; however, we encountered obstacles that prevented a clear outcome. Specifically, we prepared grids using our chameleon protocol with the oxygen sensitive dye, resazurin, that is commonly used for to detect oxygen contamination during experiments. Despite multiple attempts, we struggled to produce grids with sufficient dye signal for detection during imaging which necessitated creating thicker ice grids. This resulted in a second challenge: thicker ice tended to auto-fluoresce, introducing variability between grids that was too large to draw meaningful conclusions. We attempted to normalize the different samples but were unable to obtain consistent results.
- 5. Related to point 4, as oxidation may have to proceed to a significant degree before it becomes visible within the Hb and MoFeP structures, it may be informative to perform a series of activity measurements while the protein is kept under oil to see whether or not the proteins are losing activity even if they may not be oxidized enough to show a conformational change.**
9. For Hb, as indicated in Figure 1C, we performed an oxygen binding assay, where we monitored oxygen binding over time using UV-Vis. We see that for anaerobic deoxy Hb, in low NaDT conditions (25 μM) and without the protective oil layer, we can observe a noticeable shift in the Soret maximum over time, indicating oxygen contamination and binding to Hb. In all of the oil-layered samples we tested, no noticeable shift was observed, indicating limited oxygen perfusion through the oil layer. These experiments provide evidence that oxygen contamination from the time the sample is removed from the anaerobic vial and aspirated into the chameleon is minimal when the sample is located inside the sample cup under a protective oil layer.
10. Furthermore, oxidation of the P-cluster in MoFeP results in rearrangements of the metal clusters (Figure 3) which we do not observe in our structures when the protective oil layer and

NaDT are used. This provides strong evidence that the reduced state is the dominant, if not only, state observed in our cryoEM experiments.

6. The author's note in the discussion that they freeze 1-2 grids per sample, and that the protocol requires a higher volume of sample than is typical for the chameleon. This may be limiting for researchers with lower yield, it would be good to comment on this.

11. The SPT Labtech chameleon accommodates a range of volumes aspirated (from 3 to 10 μL) per freezing session. Each grid made during this session uses approximately 5 to 10 picolitres of sample. This requires the user to generally need a minimum of 5 μl total to reliably create high quality grids. In contrast, when using a system such as a Vitrobot, each grid requires 3+ μL of sample. Therefore, this protocol uses similar, if not less, volume than traditional vitrification approaches.

7. Why is 20 mM dithionite concentration used for the nitrogenase protein, while 60 mM dithionite was found to be necessary for Hb? Comment on this.

12. We initially used 60 mM NaDT for structure determination of MoFeP, based on our results with Hb, but then decided to try and lower the concentration of NaDT to those similar to conditions used for enzymatic assays (see comment #1.1). We were able to obtain a structure of reduced MoFeP using 5 mM NaDT. We have updated the Results and Supplementary Materials accordingly to include our new structure. We speculate that the reason a lower NaDT concentration protects MoFeP is due to: 1) the metal clusters of MoFeP being buried within the protein compared to the hemes Hb which are quite solvent-exposed and 2) there are different time-scales at play for these process. Specifically, we reason that Hb will readily bind any O_2 present in solution while the process of oxygen-induced oxidation of MoFeP occurs on a much longer timescale.

8. A minor point, in the Figure 2 legend the resolution of the 60 mM DT/Hb map is listed as 2.7 Å, while in the text it is listed as 2.6 Å, and in the figure legend it is not stipulated whether or not the map shown is C1 or C2. These points should be clarified.

13. We thank the reviewer for their careful reading of the manuscript. We have updated and changed the reported resolutions due to our change to the C1 refinements for Hb.

9. Within the abstract the authors state: "Current practices for anaerobic grid preparation involve a vitrification device located in an anoxic chamber, which presents significant challenges including temperature and humidity control" – however, the anaerobic cryoEM protocols published to date (Cherrier et al., Warmack et al., Schmidt et al., 2024) utilized vitrobots within anaerobic chambers, which control temperature and humidity. This point should be reconsidered or removed from the abstract.

14. Although the reviewer is correct that most anaerobic chambers control temperature and humidity, cryoEM sample preparation typically requires the sample to be held at 4 °C in 100% humidity. These conditions are not amenable for most anaerobic chambers since most, but not all, are kept at ambient temperature at low humidity for ease of use and to protect that catalysts.

It would be challenging to operate an anaerobic chamber such as a Coy Laboratories glovebag or a Mbraun anaerobic glovebox at 4 °C and 100 % RH.

10. There appears to be a type at the end of the Figure 2 legend, “\”

Corrected.

11. Line 294: ‘alpha’ should be the symbol for alpha.

Corrected.

12. Explain the coloring scheme used for the alpha and beta subunits in Figure 2.

We have added a description of the coloring scheme to the figure 2 legend.

Reviewer #3 (Remarks to the Author):

In the present manuscript, Cook et. al. test the suitability of a blot-free vitrification device, the SPT Labtech chameleon (hereafter referred to as chameleon), for the preparation of oxygen (O₂) sensitive cryogenic electron microscopy (cryoEM) samples. Using hemoglobin as an O₂ reporter and the catalytic component of the molybdenum nitrogenase (MoFe protein) as an O₂ sensitive proof of principle sample. The authors outline a very useful workflow for the preparation of O₂ sensitive cryoEM samples with the chameleon in a regular aerobic atmosphere with a protective oil layer and in the presence of sodium dithionite, thus enabling a wider range of scientists to freeze O₂ sensitive cryoEM samples. As discussed, anaerobic preparation of cryoEM samples by today’s standards is not optimal. To prevent O₂-induced damage to the sample, a vitrification device such as a Vitrobot is placed inside an anoxic chamber, which allows for sample preparation under an anoxic atmosphere but causes problems in humidity/temperature control, handling of the samples, etc. However, despite these drawbacks the preparation of cryoEM samples under an anoxic atmosphere ensures the anaerobic sample preparation, whereas the approach presented in this paper might not.

The chameleon is operating under an aerobic atmosphere. Therefore, preparation of hemoglobin and MoFe protein samples without any precautions yielded O₂ bound hemoglobin and O₂ damaged MoFe protein, respectively. The authors introduce two measures to overcome this. First, they place an Al’s oil layer on top of the sample reservoir that prevents O₂ diffusion into the sample. Second, they add the reductant sodium dithionite (NaDT) to their samples, a well-known O₂ scavenger that is widely used to protect O₂ sensitive samples such as the nitrogenase enzyme. In my opinion, these measures are not sufficient to refer to the overall process as anaerobic, also not as (an)aerobic.

The moment the sample is aspirated by the chameleon, it is exposed to an aerobic environment. As demonstrated by the authors, the added NaDT is scavenging most of the O₂ during the sample preparation process, thus high amounts NaDT are required to keep the samples reduced (60mM and 20mM for hemoglobin and MoFe protein samples, respectively).

Hence, I would rather refer to the overall process as aerobic in the presence of high amounts of an O₂ scavenger.

1. While we agree with the reviewer that the conditions we are using are not truly anaerobic, the sample is maintained under conditions until freezing occurs where the effects of oxygen contamination are not detectable in our final structures. As the oxygen scavenger eliminates oxygen, the sample remains anaerobic, even if held aerobically, until the concentration of NaDT sufficiently drops to allow for the dissolved oxygen concentration to increase. Although not truly anaerobic by definition, we reason that the conditions are similar to conditions maintained in most anaerobic chambers. Indeed, human Hb binds oxygen around 100 ppm, which means the oxygen concentration is lower than that in our deoxy Hb structure. We have further clarified this in the manuscript.
2. To address the concern about high NaDT concentrations, we obtained a ~2.2 Å reduced MoFeP structure in the presence of 5 mM NaDT. This concentration is in line with NaDT concentrations used in numerous other biochemical and structural studies.

Overall, the manuscript is well written and the authors present their data in a convincing manner. However, several concerns need to be addressed prior to publication:

1. My main concern regards the expression “(an)aerobic blot free vitrification”. The word “(an)aerobic” is problematic as discussed above and should be deleted in the title and the abstract. However, “(an)aerobic” could potentially be used throughout the text after the explanations in line 73f.

Moreover, I am wondering, whether it would be possible to conduct the whole process under an anoxic atmosphere? Judging by the demonstration video of the device that is available online, the vitrification process does operate in a chamber. Could one maybe modify the device setup in a way that the chamber is constantly flushed with argon or nitrogen? Making the whole process really anaerobic and with this offering a brilliant solution for O₂ sensitive cryoEM samples.

3. We thank the reviewer for their comments. One goal for this protocol is for any user to be able to use any available chameleon without having to modify the chameleon itself. This is because we envision most users to use the chameleon within a national center or university core. By having our samples prepared the way they are, there is no major modification needing to be made to the chameleon. Unfortunately, the chameleon chassis is not sealed and therefore constant flushing with argon would potentially generate an asphyxiation concern for the user of the instrument. Furthermore, the chamber that contains the grid is humidified and flushing with argon would decrease the humidity to levels that would lead to overwicking and dehydration of the samples, at least as it is currently operated. Perhaps in another generation of the instrument the chamber could be modified to accommodate such a flushing device.

2. The authors state in the abstract (line 27), the introduction (line 68) and later in the main text that the conventional way of preparing cryoEM grids anaerobically (Vitrobot inside an anaerobic chamber) is costly and not accessible to the community. In line with this argumentation, the authors present their new workflow as a more cost-effective solution and argue that the chameleon is already widely spread across cryoEM facilities and thus more

accessible. However, I am missing a direct cost comparison of the two approaches. What's the cost for purchasing and operating the chameleon? How does that compare to the conventional setup of a Vitrobot inside an anoxic chamber? Approximately how many chameleons are operating globally and can this potentially be compared to the number of Vitrobots or anoxic chambers? At least the price of the chameleon and vitrobot inside the tent should be compared, if so much emphasize is put on the cost advantage.

4. We thank the reviewer for their careful reading of our manuscript. We will update our discussion to include a cost comparison. A new Vitrobot to locate into an anaerobic chamber is ~\$100,000 plus the cost of the anaerobic chamber and location within the lab. The SPT Labtech chameleon is about \$400,000 but is typically purchased by a cryoEM core facility or a national cryoEM center (e.g., S2C2) and the user is charged for their time on it or can be made available through a proposal submission process, respectively. Thus, for a general user using a facility chameleon, the entry cost to creating anaerobic grids would be the cost of the grids and the hour of chameleon time. To prepare these same grids in an anaerobic chamber with a Vitrobot would require the lab to purchase the Vitrobot to locate into the chamber. The goal of this protocol is for users who have access to anaerobic chambers but not the funds to purchase a Vitrobot to be able to create high quality grids in core facilities and national centers.

3. Fig. 3: I assume that the chameleon device is more expensive than a Vitrobot. Given that the chameleon is also operating under an aerobic atmosphere, I am wondering if the same results obtained with the chameleon could be obtained with a Vitrobot operating under aerobic conditions, using the same O₂ protection measures established for the chameleon (20mM NaDT and a protective oil layer). Is it possible to obtain high-resolution structures of the reduced MoFe protein under these conditions with a Vitrobot? This would circumvent all problems described for operating a Vitrobot inside an anoxic chamber. Moreover, Vitrobots are the standard vitrification device and thus the most accessible. Demonstrating that this workflow would also work for a Vitrobot would significantly increase the impact of this manuscript.

5. Unfortunately, this experiment would not work as the instruments currently operate. Once the sample is applied to an EM grid, it cannot be layered with oil in the same manner as the sample located in the chameleon cup. The other concern is the time the sample would be exposed to oxygen compared to the chameleon. Specifically, once the sample is applied to an EM grid in the Vitrobot, it requires 3-10 s of blotting before being plunge frozen. In contrast, when the sample is aspirated into the chameleon dispenser, the sample is then only fully exposed to oxygen during the wicking process, which typically takes 300-600 ms. The difference in timescales means the NaDT would most likely be completely exhausted before grid vitrification using the Vitrobot.

6. Indeed, in a previous nitrogenase study, anaerobically-prepared, reduced MoFe protein was frozen using a manually-operated plunge freezer (aerobic, 4 °C, 95% humidity) in the presence of 5 mM NaDT and the oxidized form of the protein was determined, indicating that the NaDT was exhausted and the metal clusters were oxidized upon air exposure (DOI: 10.1126/science.abq764). This is in contrast to our current study, where 5 mM NaDT is sufficient to obtain reduced MoFe protein using the chameleon.

4. Line 92: The authors point out that the chameleon might be able to help with preferential orientation problems. It is known that the MoFe protein is prone to interact with the air-water interface during cryoEM sample preparation and is thus prone to preferential orientation issues (Warmack, R. A., & Rees, D. C. (2022). Anaerobic single particle cryoEM of nitrogenase. bioRxiv, 2022.2006.2004.494841. doi:10.1101/2022.06.04.494841). As described in the materials and methods (line 410) the authors used a detergent cocktail in their sample preparation for the MoFe protein structures presented in Fig. 3. Detergents are commonly used in cryoEM to alleviate the effects of preferential orientation, thus it is not clear whether the chameleon helped to overcome this challenge or the detergent cocktail. To address this question: Is it possible to get a high-resolution structure of the MoFe protein without the use of detergent when using the chameleon for sample preparation?

7. As the reviewer commented, MoFe protein is prone to air-water interface issues and partial denaturation. Unfortunately, in our experience, despite significant efforts, the chameleon alone was not able to rectify this issue (unpublished data) and preferred orientation issues persisted.

Minor comments:

• Please add the missing “S” to Table 1 and Table 2 when referencing Table S1 and S2.

Corrected.

• Line 192: “...predicted concentration of O₂ (250μM) in solutions...”. I am missing a reference for this statement.

Corrected.

• Line 277: “... we decided to use MoFeP to determine if our anaerobic freezing protocol...”. Brackets for (an)aerobic missing.

Corrected.

• Line 249: “Although the overall tetramer structure was once again R state, our analysis of the heme groups in each subunit revealed an absence of any density stretching above the Fe, indicative of O₂ binding within the hemes.”

Please rephrase to clarify, for example: “...any density stretching above the Fe, which would have been indicative of O₂ binding within the hemes.”

Corrected.

• Line 340: “Any oil remaining on the outside of the dispenser can affect sample deposition and therefore needs to be removed”. From the results described in line 138 to 163 I concluded that the oil does not adversely affect the cryoEM samples. Were “additional dispenser wash steps to remove any residual oil” (line 142) applied in these experiments? If yes, I would recommend to clarify that this is an essential step in the chameleon process.

While the oil will not disrupt the sample itself, the oil can affect the operation of the chameleon. The multiple wash steps is to ensure proper chameleon operation, not to effect the sample. The text has been adjusted to empathize this.

• Line 405: Please describe the procedure for the iron chelation assay or provide a reference.

Corrected.

- **Line 527: “-“ missing between “NU” and “refinement”.**
Corrected.

Reviewer #4 (Remarks to the Author):

REVIEWERS' COMMENTS

Reviewer #1 (Remarks to the Author):

In this revised version of their manuscript, the authors have addressed all my previous concerns. As highlighted in my earlier review, this manuscript provides a clear and detailed description of the methods employed, making it a valuable resource for the scientific community. By presenting an approach that enables structural characterization of metalloproteins at a reduced cost, this work has the potential to significantly benefit a wide range of research groups.

However, as the authors themselves acknowledge, the implementation of such a device is currently concentrated in facilities predominantly located in the United States. This geographical concentration may limit access to this method for researchers outside these regions due to the high cost of the equipment.

In addition, it is worth noting that integrating a grid vitrification device within an anaerobic chamber, while more complex and expensive, could offer exciting opportunities for studying specific redox states in metalloproteins. While probably confined to specialized research groups, this alternative approach expands the possibilities for structural characterization. Having said this, I am convinced that this manuscript represents a substantial contribution to the field and I fully support its publication in Nature Communications.

Reviewer #2 (Remarks to the Author):

I thank the authors for their important contributions in making the community aware of these important resources that are valuable to explore for the growing field of cryoEM. However, for the benefit of the community, I urge the authors to be balanced and forthcoming in their discussion of the limitations of their approach.

As my fellow reviewers have noted, certain facts are indisputable about the work as presented:

(1) The process of using a chameleon for cryoEM sample preparation is not shown here to be truly anaerobic, as demonstrated by the oxidation of the Hb when lower concentrations of reductant are added. Therefore, I strongly encourage the authors to replace the term "(an)aerobic" in their title, as it is not germane to the community and is ill defined in this context.

- We have updated the title of the manuscript to: "Preparation of oxygen-sensitive proteins for high-resolution cryoEM structure determination using blot-free vitrification"

(2) The availability of semi-automatic freezing robots (GP2 and Vitrobot) exceeds that of the

newer and more advanced Chameleons by at least an order of magnitude. While they may not be in anaerobic chambers, **the authors should test their methodology with this more canonical equipment. The samples could easily be overlaid with oil, and blotting times of less than a second can be achieved.** As we see from the data, the speed of oxygen diffusion across the thin layer of liquid created by a printing robot would exceed the time delay before freezing, making the argument against the comparison to slower freezing methods moot.

- We have attempted preliminary experiments using our oil layering method with a standard blot and plunge method (both manual freezing and vitrobot) and have found that the time scales and mechanics of the blot and plunge method do not allow for the creation of high-quality grids using our method. We have added a line within the discussion indicating this: “Although we have demonstrated this protocol to be effective for the chameleon, many aspects, including the layering of protective oil, are not currently amenable to traditional blot-and-plunge freezing protocols.” (Line 401-403).

It is true that a vitrobot in an anaerobic chamber is not ideally suited for routine use in a user facility, and perhaps some variation of the proposed protocol herein is better. However, that may require additional optimization to limit oxygen exposure, improve sample exchange, etc.

- There have been other labs that have reported methods to freeze anaerobic proteins within an anaerobic environment (Warmack et al, Nature Protocols, 2024). Our method provides an alternative which can allow a larger range of researchers access to these difficult proteins.

Reviewer #3 (Remarks to the Author):

Thank you for revising the manuscript and addressing my feedback and concerns. While I do believe the manuscript has improved significantly, I would like to highlight some points that were only insufficiently addressed. The respective points are according to the numbering of the authors:

1. Regarding the term (an)aerobic used in the title and the abstract: I have no doubt that the conditions used in the study can prevent oxygen-induced damage to oxygen sensitive samples. Moreover, I fully agree with the authors that the conditions are “not truly anaerobic by definition”, therefore I am convinced that the term “(an)aerobic” should be used carefully. In fact, I find it misleading that such a new and undefined expression is used without explanation in the title and abstract of the paper. **Thus, I request to rephrase the title and introduce the term “(an)aerobic” together with the explanation provided in line 75 in the abstract.** For the title I suggest “Preparation of reduced oxygen-sensitive proteins for high-resolution cryoEM structure determination using blot-free vitrification under air”. I am convinced that this title would be more clear and more interesting for a

general readership.

- As mentioned above, we have revised the title of our paper and removed “(an)aerobic from the abstract to remove any confusion. We have the term defined early in the intro to avoid any confusion.

4. Thank you for providing a cost comparison of the presented method with the use of a Vitrobot in a tent in the rebuttal letter. However, I am missing this comparison in the discussion of the current manuscript version, as promised in the rebuttal letter. Please add the cost comparison to the manuscript. Moreover, **please add to Table S3 also the location (country and city) not only the name of the institute.**

- We have updated TableS3 to include the state and country for each chameleon. We also have excluded the direct price comparison within the manuscript due to each chameleon and vitrobot being uniquely priced based on contracts and quotes with the prospective institution. The inclusion of the price would be misleading and distracts from the overall method presented here.

7. As the authors confirmed in the answer to my comment, the detergent cocktail is essential for overcoming the air-water interface issues of nitrogenase samples. Thus, the **authors should explicitly state that the detergent cocktail is essential for the nitrogenase samples in the manuscript, for example in the paragraph from lines 289 – 309.**

- The detergent cocktail is only necessary for the nitrogenase protein, as the Hb samples did not require it. This cocktail is necessary for the accurate reconstruction of nitrogenase as outlined in (Narehood et al., Nature, 2025) and is not a necessary component for the success of this method.

L. 320, correct “determinized” to “determined”

- Fixed